# Gut microbiota and fermentation-derived branched chain hydroxy acids mediate health benefits of yogurt consumption in obese mice

Noëmie Daniel[1,2,7], Renato Tadeu Nachbar[1,2,7], Thi Thu Trang Tran[3], Adia Ouellette[1,2], Thibault Vincent Varin[1,2], Aurélie Cotillard[3], Laurent Quinquis[3], Andréanne Gagné [1], Philippe St-Pierre[1,2], Jocelyn Trottier[4], Bruno Marcotte[1,2], Marion Poirel[5], Mathilde Saccareau[6], Marie-Julie Dubois[1,2], Philippe Joubert[1], Olivier Barbier[4], Hana Koutnikova [3✉] & André Marette [1,2✉]

Meta-analyses suggest that yogurt consumption reduces type 2 diabetes incidence in humans, but the molecular basis of these observations remains unknown. Here we show that dietary yogurt intake preserves whole-body glucose homeostasis and prevents hepatic insulin resistance and liver steatosis in a dietary mouse model of obesity-linked type 2 diabetes. Fecal microbiota transplantation studies reveal that these effects are partly linked to the gut microbiota. We further show that yogurt intake impacts the hepatic metabolome, notably maintaining the levels of branched chain hydroxy acids (BCHA) which correlate with improved metabolic parameters. These metabolites are generated upon milk fermentation and concentrated in yogurt. Remarkably, diet-induced obesity reduces plasma and tissue BCHA levels, and this is partly prevented by dietary yogurt intake. We further show that BCHA improve insulin action on glucose metabolism in liver and muscle cells, identifying BCHA as cell-autonomous metabolic regulators and potential mediators of yogurt's health effects.

[1] Quebec Heart and Lung Institute (IUCPQ), Laval University, Québec, Canada. [2] Institute of Nutrition and Functional Foods (INAF), Laval University, Québec, Canada. [3] Danone Nutricia Research, Palaiseau, France. [4] Laboratory of molecular pharmacology, CHU of Québec Research Center and Faculty of Pharmacy, Québec, Canada. [5] IT&M Innovation on behalf of Danone Nutricia Research, Neuilly-sur-Seine, France. [6] Soladis on behalf on Danone Nutricia Research, Paris, France. [7] These authors contributed equally: Noëmie Daniel, Renato Tadeu Nachbar. ✉email: hana.koutnikova-rousselin@danone.com; andre.marette@criucpq.ulaval.ca

The incidence of type 2 diabetes (T2D) is rising to epidemic proportions with recent worldwide figures reaching over 463 million people[1]. Poor dietary and lifestyle habits are implicated in T2D development and understanding the contribution of individual food groups is of public health relevance. Dairy products have been suggested to reduce T2D risk[2,3]. Among dairy products, yogurt has received a particular interest since a growing number of prospective studies have reported that yogurt intake is associated with reduced body weight gain[4–6], reduced non-alcoholic fatty liver disease (NAFLD)[7], and decreased T2D incidence[2,8–15]. Meta-analyses of prospective cohort studies also reported that yogurt consumption is inversely associated with T2D[16–18]. Moreover, a recent human intervention study showed that 24-weeks consumption of yogurt (220 g/day) improved insulin resistance and reduced liver fat in obese women with NAFLD[19]. However, the physiological and molecular mechanisms underlying the beneficial effects of yogurt consumption remain unknown.

It has been proposed that yogurt intake might reduce T2D risk through its nutrient-rich profile[3,20] and/or by the presence of specific compounds derived from milk fermentation by *Streptococcus salivarius* subsp. *thermophilus (S. thermophilus)* and *Lactobacillus delbrueckii subsp. bulgaricus* (*L. bulgaricus*) starters or the bacteria itself[3,7,21]. Whether these effects are related to changes in the composition of the gut microbiota, metabolic inflammation, and/or linked to stimulation of specific metabolic pathways remain to be explored.

The aim of this study was therefore to determine the effect of yogurt intake on the development of whole-body and tissue-specific insulin resistance in a mouse model of obesity-linked T2D and to determine the potential role of the gut microbiota and hepatic metabolic pathways. We found that lyophilized yogurt intake (equivalent to two servings of yogurt) reduced the development of high-fat diet-induced insulin resistance, which was linked to its effects on liver metabolism and the gut microbiota. These beneficial metabolic effects of yogurt intake were associated with an increase of branched-chain hydroxy acids (BCHA) in the liver of these animals. We further show that BCHA are specifically present in yogurt and generated upon milk fermentation. Remarkably, BCHA modulated glucose metabolism in both hepatocytes and myocytes in vitro providing evidence for a cell-autonomous control of liver and muscle metabolism by these yogurt-specific metabolites. Thus, our work identifies BCHA as new milk fermentation-derived molecules that potentially contribute to the beneficial effects of yogurt intake on T2D and related metabolic disturbances.

## Results

**Yogurt consumption preserves glucose homeostasis and insulin sensitivity.** Three individual studies were performed to explore the effect of a lyophilized yogurt product (Y) on diet-induced obesity and associated metabolic impairments using a high-fat high-sucrose (H) diet whose protein source was not limited to casein but was a mixed source of protein to be more representative of those consumed by humans[22] (Table 1). The yogurt was administered in the lyophilized form to reach the equivalent of two servings per day, hence replacing 7.6% of the daily energy intake. The lyophilized yogurt was incorporated in the H diet every day and all the diets were changed daily. The first two studies (Study 1, *n* = 18–24 per group and Study 2, *n* = 14–24 per group) were carried out to explore the overall impact of yogurt treatment on body weight, hepatic metabolism, and indices of glucose homeostasis and insulin resistance after 12 and 15 weeks of dietary treatments, respectively. The third study (Study 3, *n* = 36 per group) was designed to investigate the impact of yogurt

treatment on the whole body, hepatic and peripheral insulin sensitivity using the gold standard hyperinsulinemic-euglycemic clamp (HIEC) method. The experimental designs are detailed in Supplementary Fig. 1, and parameters measured in the three studies are recapitulated in Supplementary Data 1.

As expected, 12 weeks of H diet feeding induced obesity when compared to the low-fat low-sucrose control diet (C) in all three studies (Fig. 1a and Supplementary Fig. 2A, B). The yogurt treatment slightly and transiently reduced body weight gain in H-fed animals in the first study (from week 6 to 9) (Fig. 1a). While this was not significant in the two others (Supplementary Fig. 2A, B), a pooled analysis of the three studies showed a small 2.9% decrease in body weight of H-fed animals after 12 weeks of yogurt treatment (Fig. 1b). Interestingly, energy intake was slightly but significantly lower (*P* < 0.05) in yogurt-treated group in each of the three individual studies (Fig. 1c and Supplementary Fig. 2C, D), as also revealed by the pooled analysis (Fig. 1d). While total relative fat mass was not affected by dietary yogurt treatment, we observed a lower inguinal adipose tissue weight (iWAT) in Y-fed animals as compared to H mice (Supplementary Data 1). Yogurt treatment significantly increased relative lean mass in Study 1 (Supplementary Data 1). There was no significant effect of yogurt treatment on energy excretion, the weight of other fat depots or on the plasma lipid profile (Supplementary Data 1).

We next measured the impact of yogurt treatment on obesity-related metabolic phenotypes. Pooled analysis of the three studies showed that fasting glycemia, fasting insulinemia, and insulin resistance, estimated by the HOMA-IR calculation were lower in Y-fed mice after 12 weeks of treatment (Fig. 1e–g). Oral glucose tolerance test (oGTT) was performed at the end of Study 1 and revealed that Y-treated mice displayed similar glucose excursion, but lower glucose-stimulated insulin response (GSIS) as compared to H-fed mice, further indicating a higher insulin sensitivity after yogurt treatment (Fig. 1h, i). We also observed a numerical decrease in C-peptide levels during the oGTT (Fig. 1j), suggesting that lower GSIS may be related to decreased insulin secretion by the beta-cells. Yogurt treatment also modestly but not significantly displayed better fasting insulin, HOMA-IR, and GSIS after 15 weeks of treatment of H-fed mice (Supplementary Fig. 2E–I) suggesting that a greater sample size rather than longer duration of treatment may be necessary to detect a significant effect. Accordingly, increasing the number of animals from 24 to 36 in Study 3 allowed us to detect a significant effect of yogurt on fasting glycemia and insulinemia (Supplementary Fig. 2G–I). We also performed tracer-coupled HIEC studies in this latter group of mice to thoroughly examine whole-body insulin sensitivity and determine the potential site(s) of better insulin action in yogurt-treated H-fed mice. These results confirmed that H-fed mice were extremely resistant to insulin as revealed by the marked decrease in whole-body glucose infusion rate (GIR) during insulin infusion when compared to C-fed mice (Fig. 1k). This was linked to both reduced insulin suppression of hepatic glucose production (Ra) and lack of insulin-induced glucose disposal (Rd) in H-fed clamped animals (Fig. 1l, m). Yogurt treatment significantly prevented insulin resistance of H-fed mice as shown by higher GIR values in the Y group (*p* < 0.05), which was associated with a trend for higher insulin-suppressed hepatic glucose production (Ra) (*p* = 0.06) (Fig. 1l). Although yogurt-treated H-fed mice did not show higher Rd values as compared to their H-fed counterparts during the clamp, it is important to note that yogurt treatment significantly preserved insulin response as compared to basal Rd values (Fig. 1m).

**Yogurt consumption prevents hepatic steatosis and dysfunction.** Dietary yogurt treatment for 12 weeks prevented H-fed

**Table 1 Diet composition.**

| Macronutrient | C | | | H | | | Y | | |
|---|---|---|---|---|---|---|---|---|---|
| | Study 1 | Study 2 | Study 3 | Study 1 | Study 2 | Study 3 | Study 1 | Study 2 | Study 3 |
| | % kcal | | | % kcal | | | % kcal | | |
| Protein | 19.2 | 19.2 | 19.2 | 15.1 | 15.0 | 15.1 | 15.0 | 15.0 | 15.0 |
| Carbohydrate | 68.1 | 68.0 | 68.0 | 34.9 | 35.0 | 35.0 | 35.0 | 35.0 | 35.0 |
| Fat | 12.7 | 12.8 | 12.8 | 50.0 | 50.0 | 50.0 | 50.0 | 50.0 | 50.0 |
| SAT:PUFA ratio | 1.0 | 1.3 | 0.9 | 0.9 | 1.3 | 0.9 | 0.9 | 1.3 | 0.9 |
| kcal/g | 3.7 | 3.7 | 3.7 | 4.8 | 4.8 | 4.8 | 4.8 | 4.8 | 4.8 |
| **Ingredient** | **g** | | | **g** | | | **g** | | |
| Lyophilized yogurt product | 0.0 | 0.0 | 0.0 | 0.0 | 0.0 | 0.0 | 8.3 | 8.5 | 9.1 |
| Protein | 0.0 | 0.0 | 0.0 | 0.0 | 0.0 | 0.0 | 2.5 | 2.8 | 2.8 |
| Carbohydrates | 0.0 | 0.0 | 0.0 | 0.0 | 0.0 | 0.0 | 3.7 | 3.7 | 4.1 |
| Fat | 0.0 | 0.0 | 0.0 | 0.0 | 0.0 | 0.0 | 1.0 | 1.2 | 1.1 |
| Protein mix[a] | 21.0 | 19.6 | 19.7 | 22.0 | 20.4 | 20.6 | 18.7 | 17.0 | 17.1 |
| Protein | 17.2 | 17.3 | 17.2 | 18.0 | 17.9 | 18.0 | 15.3 | 15.0 | 15.0 |
| Carbohydrates | 0.1 | 0.2 | 0.0 | 0.1 | 0.2 | 0.0 | 0.1 | 0.2 | 0.0 |
| Fat | 2.7 | 1.2 | 1.4 | 2.8 | 1.2 | 1.5 | 2.4 | 1.0 | 1.2 |
| L-cystine | 0.3 | 0.3 | 0.3 | 0.2 | 0.2 | 0.2 | 0.2 | 0.2 | 0.2 |
| Corn starch[b] | 56.9 | 56.8 | 56.9 | 6.5 | 6.5 | 6.5 | 6.6 | 6.5 | 6.5 |
| Sucrose | 9.0 | 9.0 | 9.0 | 34.0 | 34.0 | 34.1 | 29.6 | 30.1 | 29.6 |
| Cellulose (fiber) | 5.0 | 5.0 | 5.0 | 5.0 | 5.0 | 5.0 | 5.0 | 5.0 | 5.0 |
| Lard | 0.5 | 2.9 | 1.7 | 13.5 | 19.4 | 15.0 | 13.1 | 17.5 | 13.0 |
| Corn oil | 2.0 | 1.1 | 2.1 | 10.5 | 6.2 | 10.3 | 10.1 | 6.9 | 11.1 |
| Mineral mix[c] | 3.5 | 3.5 | 3.5 | 6.7 | 6.7 | 6.7 | 6.8 | 6.7 | 6.7 |
| Vitamin mix[d] | 1.5 | 1.5 | 1.5 | 1.4 | 1.4 | 1.4 | 1.4 | 1.4 | 1.4 |
| Choline bitartrate | 0.3 | 0.3 | 0.3 | 0.2 | 0.2 | 0.2 | 0.2 | 0.2 | 0.2 |
| BHT[e] | 0.03 | 0.03 | 0.03 | 0.03 | 0.03 | 0.03 | 0.03 | 0.03 | 0.03 |
| Total | 100.0 | 100.0 | 100.0 | 100.0 | 100.0 | 100.0 | 100.0 | 100.0 | 100.0 |

[a]Home-made protein mix, formulated with animal (26.6% beef, 23.1% pork, 22.0% chicken, 6.8% white egg, 10.3% casein) and vegetal (11.2% soy) proteins.
[b]MP Biomedical corn starch: 90% starch, 10% humidity.
[c]MP Biomedical mineral mixture 76–12% of sucrose.
[d]Tecklad Vitamin mix AIN-76A—98% of sucrose.
[e]Tert-butylhydroxytoluene.

induced liver steatosis in Study 1 as revealed by significantly lower relative liver weight and hepatic triglyceride levels (Fig. 2a, b), which was also apparent from histological examination of the livers (Fig. 2c). Qualitative histopathological analyses of liver sections showed that yogurt-treated animals displayed a generally healthier liver phenotype as revealed by reduced hepatic steatosis grade (notably grades 2 and 3) and diagnosis and fibrosis grading as compared to untreated H-fed mice (Fig. 2d–f).

Next, we explored the hepatic expression of genes involved in fatty acid uptake, de novo lipogenesis, and fatty acid oxidation. We found that livers of mice treated with yogurt in Study 1 displayed lower expression of *Scd1* coding for the rate-limiting enzyme involved in the synthesis of monounsaturated fatty acids from saturated fatty acids and triglycerides (Fig. 2g), as well as decreased expression of MCAD (*Acadm*) and ACOX1 (*Acox1*) genes involved in both mitochondrial and peroxisomal fatty acid beta-oxidation (Fig. 2g). Yogurt treatment also tended to show reduced CD36 (*Cd36*, $p = 0.07$) and ACCα (*ACACA*, $p = 0.07$) gene expression (Fig. 2g) while SREBP1 (*Srebf1*), CPT1A (*CPT1A*) and FAS (*Fasn*) were not changed (Supplementary Fig. 3A). Multiplex analyses further suggested that yogurt intake blunted hepatic inflammation as shown by small but consistent reduction of several chemokines and cytokines in the liver of yogurt-treated H-fed animals although these effects failed to reach statistical significance (Supplementary Fig. 3B). Taken together with the clamp data, these findings demonstrate that the beneficial effects of yogurt intake on glucose homeostasis and insulin resistance are linked to a healthier liver metabolism in H-fed mice.

Prolonging dietary yogurt treatment for 15 weeks in Study 2 did not further prevent liver steatosis (Supplementary Fig. 3G, H) and did not prevent H-induced changes in hepatic gene expression and inflammation (Supplementary Fig. 3I, J) but it should be noted that these mice were fasted for 6 h instead of 4 h in Study 1 which resulted in higher TG levels, which may explain the lack of significant yogurt effect in this setting.

**Role of the gut microbiota in the beneficial effect of yogurt intake**. We next explored whether the metabolic effects of yogurt intake could be linked to changes in the gut microbiota. Yogurt treatment of H-fed mice was associated with inconsistent changes in α-diversity in between studies (Fig. 3a). Dietary yogurt treatment was also globally associated with a shift of β-diversity in H-fed mice in Study 1 as depicted by the Bray-Curtis dissimilarity ($p = 0.03$) and unweighted UniFrac distance ($p = 0.04$) (Fig. 3b). The DESeq log 2 fold changes analysis at the genus level revealed that yogurt treatment led to an enrichment of *Streptococcus* and a depletion of an unknown *Peptostreptococcaceae* genus relative abundance in the fecal microbiota of mice from all studies compared to untreated H-fed controls (Fig. 3c). The *S. thermophilus* CNCM I-1630 was quantified in fecal material and total cell counts reached 9 $\log_{10}$/g in both feces and caecum after yogurt feeding (Supplementary Fig. 4).

The effect of yogurt intake on the fecal microbiota composition was not associated with a reduction in metabolic endotoxemia as revealed by the lack of changes in either LPS or LBP circulating levels (Supplementary Data 1). Interestingly, yogurt treatment

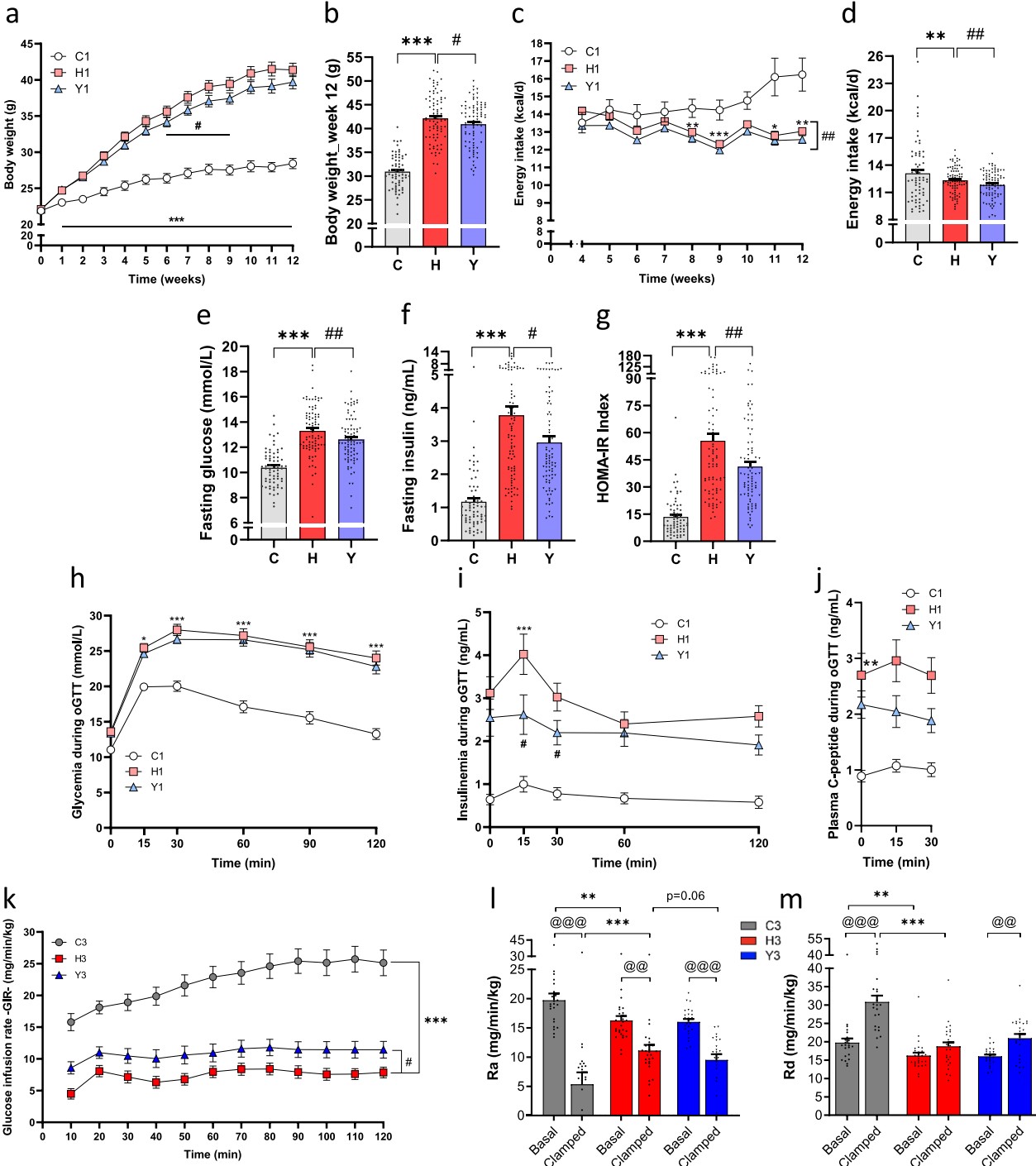

**Fig. 1 Yogurt consumption preserves glucose homeostasis and insulin sensitivity.** Metabolic results of mice in the three studies (see also Supplementary Fig. 1 for the experimental design, Supplementary Data 1 and Supplementary Fig. 2). **a** Body weight throughout the Study 1. **b** Pooled analysis of the three studies showing mice final body weight after 12 weeks of dietary treatment. **c** Energy intake was recorded from week 4 to week 12 in Study 1. **d** Pooled analysis of the three studies showing mice daily energy intake. **e**–**g** Pooled analysis of fasting glucose, insulin, and corresponding HOMA-IR index after 12 weeks of treatment. **h** Glucose tolerance test at week 11 in Study 1. **i** Insulin response following the oral glucose challenge in Study 1. **j** C-peptide response following the oral glucose challenge in Study 1. **k**–**m** In Study 3, mice underwent an hyperinsulinemic-euglycemic clamp at week 12. **k** Glucose infusion rate (GIR). **l** Glucose rate of appearance (Ra) depicting hepatic glucose production. **m** Whole-body glucose rate disappearance (Rd) depicting glucose disposal. Data are expressed as mean ± SEM. Study 1: $n = 18$–24 except for panel (**i**) ($n = 9$–12) and (**j**) ($n = 6$–10); Study 3: $n = 23$–27; Pooled analysis: $n = 66$–84 biologically independent mice. H versus C: $*p < 0.05$, $**p < 0.01$, $***p < 0.001$. Y versus H: $\#p < 0.05$, $\#\#p < 0.01$. For clamps data, basal versus clamped condition: $@@p < 0.01$, $@@@p < 0.001$. C: low-fat low-sucrose control diet (C: soft gray, C1: white, C3: gray); H: high-fat high-sucrose diet with a protein mixture replacing casein (H: soft red, H1: pink, H3: red); Y: lyophilized yogurt incorporated in H diet (Y: soft blue, Y1: light blue, Y3: blue). Numbers (1, 2, 3) refer to the study affiliation. Two-way repeated-measures ANOVA or Mann–Whitney tests depending on data distributions with (**a**, **h**–**j**) or without (**b**–**g**, **k**–**m**) baseline adjustment. $T$-test at baseline (**j**). All tests were two-sided.

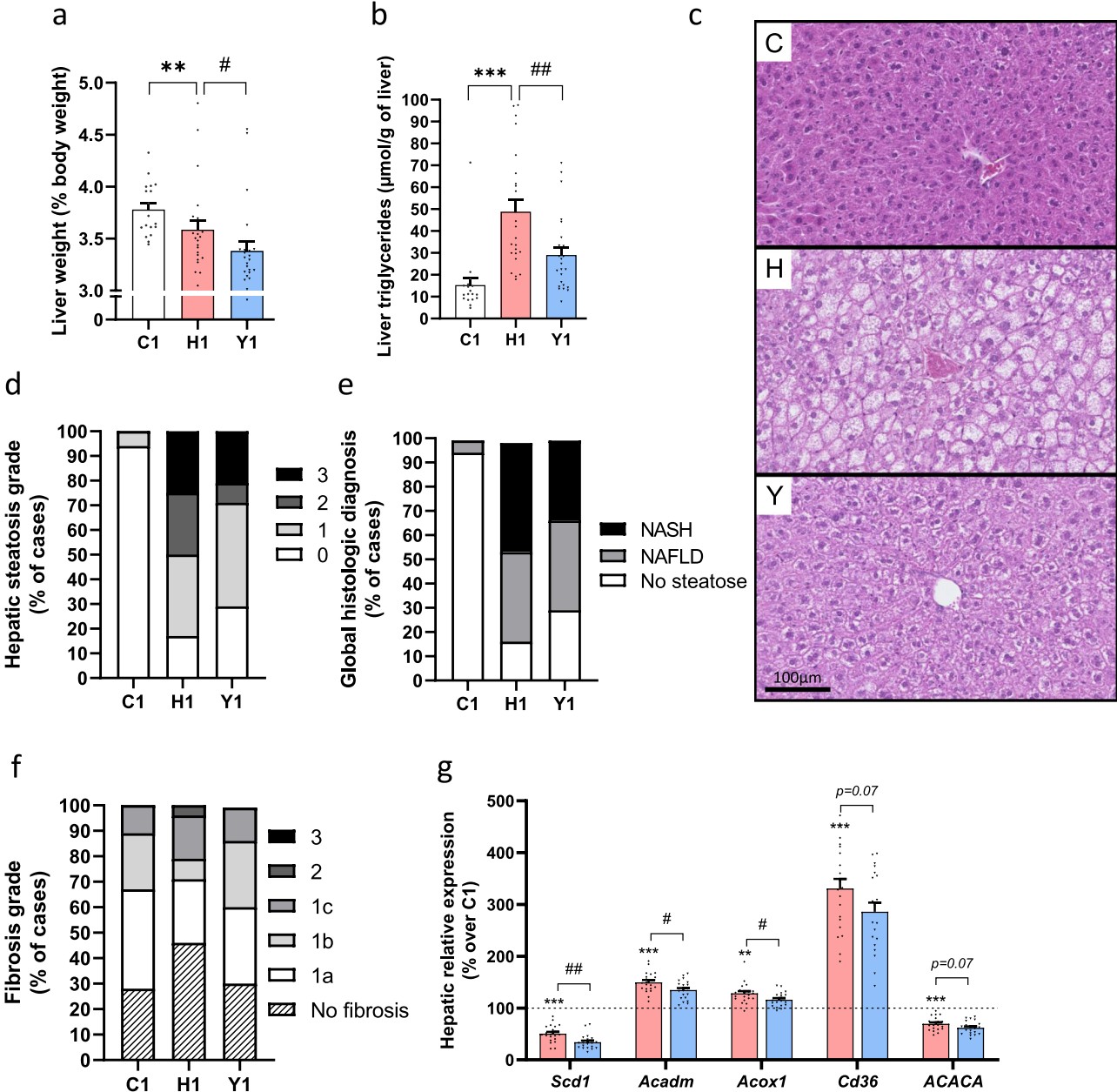

**Fig. 2 Yogurt preserves hepatic steatosis and function.** Liver state in mice from Study 1. **a** Liver weight. **b** Hepatic triglyceride content. **c** Representative liver sections (hematoxylin and eosin staining) of $n = 16$–24 sections from independent mice. **d**, **e** Pathological analyses of liver sections. **d** Hepatic steatosis. **e** Global histologic diagnosis. **f** Fibrosis score. **g** Liver expression of genes involved in fatty acid uptake, de novo lipogenesis and fatty acid oxidation. Data are expressed as mean ± SEM. $n = 18$–24 biologically independent mice, except for panel (**g**) ($n = 8$ for C1 group, $n = 20$ for H1 and Y1 groups). H versus C: $**p < 0.01$, $***p < 0.001$. Y versus H: $\#p < 0.05$, $\#\#p < 0.01$. For panel (**g**) expressed in percentage, C group is considered as a reference and represented by a dash line. C: low-fat low-sucrose control diet (C1: white); H: high-fat high-sucrose diet with a protein mixture replacing casein (H1: pink); Y: lyophilized yogurt incorporated in H diet (Y1: light blue). Number (1) refers to the study affiliation. Mann–Whitney tests or $T$-test depending on data distributions (**a**, **b**). One-way ANOVAs and Benjamini–Hochberg adjustment for multiple testing on the number of variables (**g**). All tests were two-sided.

preserved claudin-3 gene expression in the ileum of H-fed animals whereas neither mucin-2 nor occludin gene expression was affected (Supplementary Fig. 3C). No effect of yogurt intake was observed on ileum inflammation- and bile acid metabolism-related gene expression (Supplementary Fig. 3D, E). However, we found an increase of the 6α-hydroxylated acid, hyodeoxycholic acid (HDCA) in fecal content of mice fed yogurt for 3 weeks in Studies 1 and 2 (Fig. 3d), and this effect was further visually amplified after 12 weeks of yogurt feeding in Study 3 indicating altered intestinal bacteria bile metabolism (Supplementary Fig. 3F

and Data 1). No differences were found in cecal short-chain fatty acid content between the Y- and H-fed groups (Supplementary Data 1).

To explore the role of the gut microbiota in the beneficial effects of yogurt treatment in H-fed mice, we next performed fecal microbiota transplantation (FMT) studies where germ-free (GF) mice were inoculated with fecal slurries recovered from H-fed or Y-treated animals from Study 1 and fed H diet, denominated as H1-T and Y1-T mice. FMT of the microbiota from yogurt-treated mice reproduced the healthier metabolic

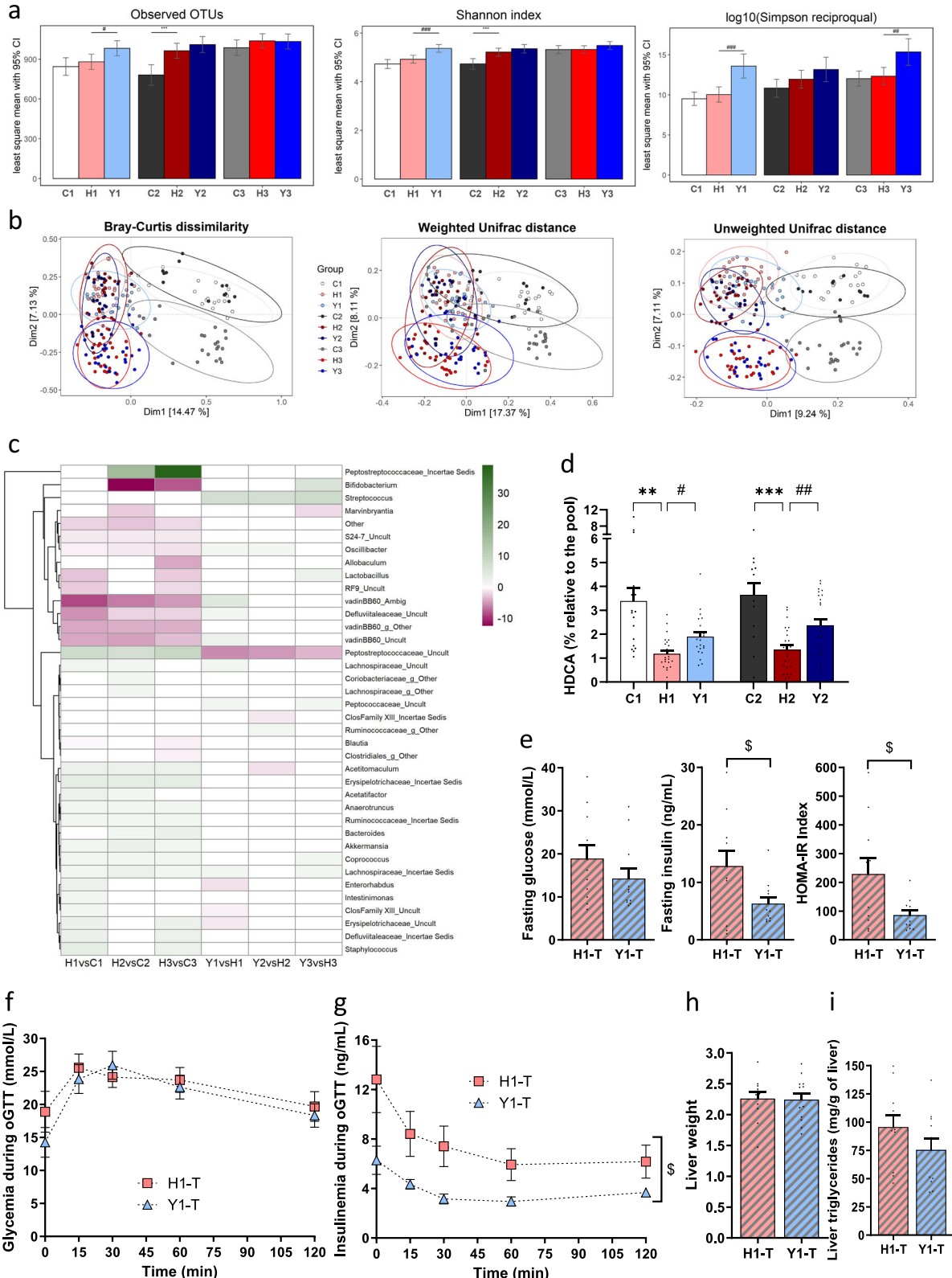

phenotypes observed in animals consuming yogurt compared to controls submitted to FMT from H-fed donors as revealed by lower fasting insulinemia and HOMA-IR values ($p < 0.05$) (Fig. 3e) as well as reduced GSIS during the oGTT (Fig. 3f, g). However, no differences were observed on liver weight whereas a numerical but not significant reduction was seen on hepatic triglyceride levels (Fig. 3h, i). The beneficial effect of FMT from

donor Y1-treated mice has not observed in a group of gnotobiotic mice fed a chow diet (Supplementary Fig. 5A, B), suggesting that a H-mediated compromised metabolic trait is needed to observe the beneficial impact of the yogurt-associated fecal microbiota. Altogether, these studies are consistent with the implication of changes in the gut microbiota composition and function in the preserved insulin sensitivity of yogurt-treated mice.

**Fig. 3 Yogurt impact is partly driven by the gut microbiota. a–c** Fecal gut microbiota assessed at 12 weeks for Study 1, 3, and 15 weeks for Study 2. **a** Alpha diversity-observed OTUs, Shannon index, and Simpson reciprocal index (indices adjusted by their baseline value, least square mean and 95% CI). **b** Beta diversity-Principal Coordinate Analyses (PCoA) of Bray-Curtis dissimilarity, weighted UniFrac distances, and unweighted UniFrac distances with 95% confidence ellipses by group. **c** Heatmap of DESeq log2 fold changes over C or H group for genera significantly altered by H diet or Y treatment, respectively. Non-significant fold changes set to 0. Ward's clustering criterion applied on genera. **d** Fecal hyodeoxycholic acid content at week 3 in Studies 1 and 2. For panels (**a–d**): Study 1, $n = 18$–24; Study 2, $n = 13$–24 and Study 3, $n = 23$–27 biologically independent mice. **e–i** Germ-free mice gavaged with fecal material from Study 1 and fed H diet for 9 weeks. Three cages of $n = 3$–4 receiver mice were allocated to each group, named H1-transplanted (H1-T) and Y1-transplanted (Y1-T). At week 9 post fecal material transplantation, **e** Fasting glucose, insulin, and corresponding HOMA-IR, **f** glucose tolerance test (GTT) and **g** glucose-stimulated insulin secretion during GTT. **h** Liver weight and **i** Hepatic triglyceride content. For panels (**e–i**): $n = 11$–12 biologically independent mice. Data are expressed as mean ± SEM except if specified otherwise. H versus C: **$p < 0.01$, ***$p < 0.001$. Y versus H: #$p < 0.05$, ##$p < 0.01$, ###$p < 0.001$. Y1-T versus H1-T: $$p < 0.05$. C: low-fat low-sucrose control diet (C1: white, C2: dark gray, C3: gray); H: high-fat high-sucrose diet with a protein mixture replacing casein (H1: pink, H2: dark red, H3: red); Y: lyophilized yogurt incorporated in H diet (Y1: light blue, Y2: dark blue, Y3: blue); T: transplanted mice (H1-T: pink, hatched and Y1-T: light blue, hatched). Numbers (1, 2, 3) refer to the study affiliation. **a** Generalized least-squares model and Benjamini–Hochberg adjustment for multiple testing on the number of variables. One-way ANOVAs or Mann–Whitney tests depending on data distribution (**d**). **e**, **h**, **i** T-test or Mann–Whitney and (**f**, **g**): mixed model followed by Tukey post hoc test. All tests were two-sided.

**Metabolomics identify BCHA as common liver and yogurt metabolites.** Since 12-week yogurt treatment prevented hepatic steatosis in Study 1, we next examined the liver metabolome to explore potential mechanisms that could explain the protective effects of yogurt on liver metabolism. Based on epidemiological evidence and previous meta-analyses, we also reasoned that the beneficial effects of yogurt may come from metabolites derived from milk fermentation and therefore focused on liver metabolites that would also be selectively overrepresented in yogurt vs non-fermented milk.

We performed targeted lipidomics (CLP platform) analysis to quantify eleven lipid classes including the complete fatty acid composition of each lipid. Among the 999 lipid species detected in liver by the CLP platform, 712 were differentially abundant between the C-fed lean and H-fed obese mice and 20 of them were also differentially modulated by yogurt treatment (q-value ≤ 0.05) (Supplementary Data 2). However, none of these hepatic lipids was enriched or depleted upon yogurt fermentation when comparing the lipidomic profile of the lyophilized yogurt and milk, suggesting that the beneficial effect of yogurt is linked to another class of metabolites (Supplementary Data 3). We next performed untargeted metabolomics (HD4 platform) analysis to quantify chemically and structurally diverse metabolites involved in amino acid, carbohydrate, cofactor, vitamin, energy, lipid, nucleotide, peptide, and xenobiotics metabolism. The HD4 platform profiling of the liver metabolome showed that 227 out of the 563 metabolites identified in the liver of obese H-fed mice were significantly differently abundant as compared to lean C-fed mice (q-value ≤ 0.05) (Supplementary Data 4). In this set, no detectable metabolite was significantly different between H and Y-fed groups (q-value ≤ 0.05) but reducing the statistical threshold (p-value ≤ 0.05) allowed us to identify a total of 35 metabolites that were differently abundant in the liver of yogurt-treated H-fed mice compared to H-fed controls (Supplementary Data 4). We thus compared this set of metabolites with those significantly different between yogurt and milk products (Supplementary Data 3) and found six metabolites that were increased in yogurt compared to milk, and overrepresented in the liver of yogurt-treated H-fed mice: N-acetylglycine, ornithine, N-acetylserine, alpha-hydroxyisocaproate (HICA), 2-hydroxy-3-methylvalerate (HMVA), and alpha-hydroxyisovalerate (HIVA) (Supplementary Data 3 and 4). The last three metabolites (HICA, HMVA, and HIVA) are BCHA and were of particular interest since they were elevated in the liver of yogurt-treated mice and lean C-fed controls compared to H-fed obese mice (Fig. 4a, Supplementary Data 4). They were also markedly concentrated (by 96, 119, and 257-fold, respectively) in the yogurt product as compared to the non-fermented lyophilized milk (Fig. 4b, Supplementary Data 3).

**Levels of fermentation-derived BCHA correlate with the metabolic benefits of yogurt intake.** Following metabolomic analysis, we next used nuclear magnetic resonance (NMR) to quantify BCHA levels. This confirmed that the three BCHA are present in high quantity in the yogurt product whereas their levels were below the limit of detection in lyophilized milk (<1.2 μM for HICA and HIVA, <2.5 μM for HMVA) (Fig. 4c). HICA and HIVA were over-represented compared to HMVA, resulting in a HICA: HMVA: HIVA ratio of 1:0.5:1.

To confirm that BCHA are produced upon milk fermentation, we also quantified them in several dairy products such as non-fermented and fermented milks, non-lyophilized yogurt product used in the study, and three other yogurts, or skyr (Fig. 4d). Remarkably, we could not detect them in non-fermented milks but did find them at varying concentrations in all tested commercially available fermented products (Fig. 4d and Supplementary Fig. 6A, B). To explore the biological role of BCHA in the preventive effect of yogurt treatment, we next correlated their hepatic abundance with relevant metabolic parameters (Study 1 and 2). We found in Study 1 an inverse correlation between BCHA hepatic levels and fasting glucose (HICA $r^2 = 0.237$, $p = 0.035$; HMVA $r^2 = 0.370$, $p = 0.016$; HIVA $r^2 = 0.327$, $p = 0.016$), as well as hepatic triglyceride content (HICA $r^2 = 0.210$, $p = 0.042$; HMVA $r^2 = 0.485$, $p = 0.002$; HIVA $r^2 = 0.451$, $p = 0.002$) (Fig. 4e). This latter association was confirmed in Study 2 (Supplementary Fig. 7). Interestingly, HICA, HMVA, and HIVA fall within the metabolism of branched-chain amino acids (BCAA) (Fig. 4f) as they are derived from leucine, isoleucine, and valine, respectively. Impaired BCAA metabolism is implicated in insulin resistance and a BCAA metabolic signature has been reported to predict insulin resistance and T2D risk in humans[23–25], but the potential metabolic roles of their BCHA derivatives remain unknown. We have thus directly measured their levels by targeted LC-MS/MS in plasma, liver, and muscle tissues from mice treated with dietary yogurt for 12 weeks (Study 1). Mean plasma levels of BCHA ranged between 0.19 and 3.86 μM (in line with human range[26], and mean liver content ranged between 0.02 and 0.70 ng/mg in liver and 0.04–0.57 ng/mg in muscle tissues, with HIVA levels being about one order of magnitude higher than HMVA and HICA (Fig. 4g–i). Importantly, the data further show that HICA, HMVA, and HIVA levels are all decreased in plasma and liver of H-fed obese mice (and to a lesser extent in the muscle) as compared to C-fed lean controls. Remarkably, yogurt treatment of H-fed mice partially prevented the loss of BCHA in both the plasma and liver of H-fed mice (Fig. 4g, h) and a similar trend was observed for HIVA and HMVA in skeletal muscle (Fig. 4i). HIVA levels increased more than HICA in yogurt-treated mice although

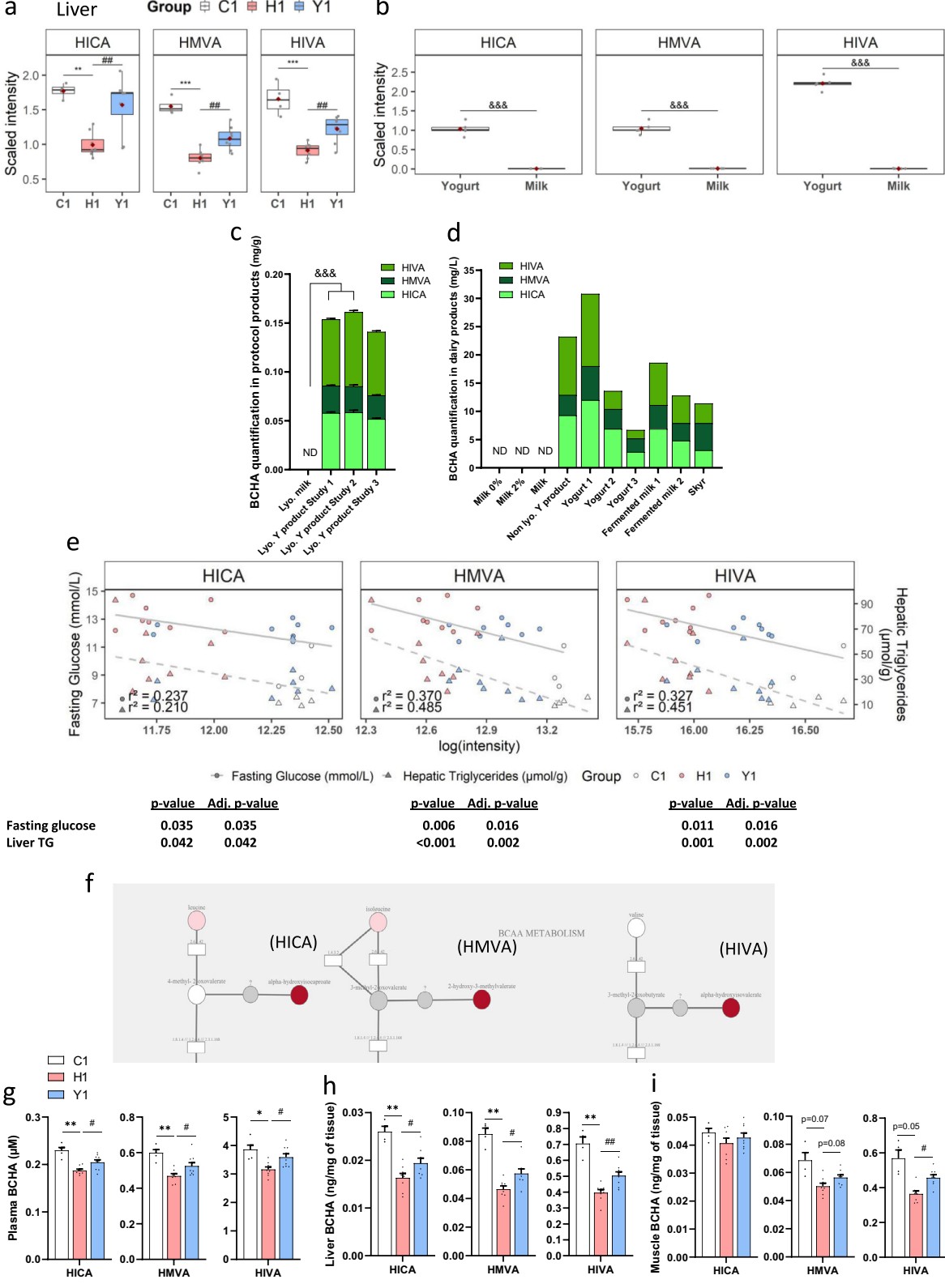

their levels were similar in the yogurt product, and that HMVA was also clearly enhanced in these animals despite representing only ~20% of the BCHA found in yogurt product, suggesting that the systemic and tissue levels of these metabolites are not only reflecting their nutritional exposure but may also be linked to some BCHA type-specific absorption and/or metabolism (Fig. 4c, g–i). These quantitative LC-MS/MS results confirm the liver metabolome data and further demonstrate that diet-induced obesity causes a dysregulation of BCAA metabolism that specifically leads to loss of BCHA production in both the circulation and in metabolic tissues, and that this defect can be partly prevented by yogurt consumption.

**Fig. 4 Yogurt intake increases levels of branched-chain hydroxy acids (BCHA) in metabolic tissues of H-fed mice which correlates with metabolic benefits. a** Hepatic BCHA levels in Study 1: alpha-hydroxyisocaproate (HICA, lawn green), 2-hydroxy-3-methylvalerate (HMVA, dark green), and alpha-hydroxyisovalerate (HIVA, green) (HD4 UPLC-MS/MS, $n = 4$–8 mice). **b** BCHA levels in lyophilized yogurt and milk (HD4 UPLC-MS/MS, $n = 5$ productions). **c** BCHA content in lyophilized milk and yogurts used in all studies (targeted NMR, $n = 5$ productions in Study 1 and 2, $n = 1$ in Study 3, average of 5 replicates). **d** BCHA content in various dairy products (targeted NMR, $n = 1$, average of 5 replicates unless indicated otherwise). **e** Pearson correlations between hepatic BCHA levels and fasting glucose (solid gray line) and hepatic triglycerides (dash gray line). $n = 4$–8. *P*-value and adjusted *p*-value indicated. **f** BCHA metabolic pathway. BCAA are metabolized by EC 2.6.1.42 to oxo-acids and then by EC 1.2.4.4; EC 1.8.1.4 and EC 2.3.1.168 complex. Circles: metabolites; rectangles: enzymes. Red: $p \leq 0.05$ between H1 and Y1; pink: $0.05 < p < 0.10$. **g–i** BCHA content in **g** plasma, **h** liver and **i** muscle (targeted LC-MS/MS, $n = 4$–5 in C and 8–9 mice in H, Y groups). Data are expressed as mean ± SEM. **a**, **b** Thick black line is median, box spans from Q1 (25th percentile) to Q3 (75th percentile), whiskers extend to the most extreme observation within 1.5 times the interquartile range (Q3–Q1) from the nearest quartile. Data are scaled such that the median value measured across all samples was set to 1.0. Outliers-dots outside the whiskers of the plot. H vs C: \**p* < 0.05, \*\**p* < 0.01, \*\*\**p* < 0.001. Y vs H: #*p* < 0.05, ##*p* < 0.01. Lyophilized milk vs lyophilized yogurts products used in studies: &&&*p* < 0.001. H1 versus C1: \**p* < 0.05, \*\**p* < 0.01, \*\*\**p* < 0.001. Y1 vs H1: #*p* < 0.05, ##*p* < 0.01. Lyophilized yogurt products vs milk: §§§*p* < 0.001. C: low-fat low-sucrose control diet (C1: white); H: high-fat high-sucrose diet with a protein mixture replacing casein (H1: pink); *L.*: *Lactobacillus*; *Lc.*: *Lactococcus*; ND: non detected; *S.*: *Streptococcus*; Y: lyophilized yogurt incorporated in H diet (Y1: light blue). Number (1) refers to the study affiliation. **a** ANOVA, **b**, **g–i** T-tests, **e** Pearson correlations. Benjamini–Hochberg correction for multiple testing within each BCHA (**e**) or within tissues (**g–i**). FDR (*q*-value) correction for multiple testing over all tested metabolites (**a**, **b**). All tests were two-sided.

---

**BCHA are cell-autonomous modulators of hepatic and muscle glucose metabolism.** Since yogurt-treated H mice displayed greater whole-body and tissue insulin sensitivity, we next questioned whether BCHA are merely a biomarker of the preventive effect of yogurt or whether they could be directly involved in these beneficial metabolic phenotypes. Given that we found BCHA to be reduced in H-fed obese mice and that this decrease was partially prevented by yogurt treatment in the liver and to a lesser extent in skeletal muscle of H-fed animals, we elected to test the hypothesis that BCHA could exert direct effects in relevant cellular models. First using FAO hepatic and L6 muscle cells, we investigated the effect of BCHA on glucose production and glucose uptake, respectively. We found that a mixture of HICA: HIVA: HMVA used at their relative ratio as in the lyophilized yogurt product dose-dependently reduced both basal glucose production and increased the suppressive effect of insulin in FAO cells (Fig. 5a, b). The increased insulin response was highly significant at 1 mM ($p < 0.001$) (Fig. 5b). Furthermore, HICA dose-dependently inhibited basal and insulin-suppressed hepatic glucose production (HGP) and competed for the effect of its BCAA leucine precursor on this metabolic process (Fig. 5c, d). Furthermore, we found that the BCHA mixture used at 0.1–1 µM concentrations to match their plasma levels in yogurt-treated mice also increased glucose uptake in L6 myocytes (Fig. 5e, f), and that these effects appeared to be mostly explained by HICA as determined from studies using individual BCHA (Fig. 5g, h). Overall, the in vitro studies provide mechanistic evidence that BCHA, and especially HICA, are cell-autonomous modulators of liver glucose production and muscle glucose uptake which most likely contributed to the beneficial effects of yogurt treatment in H-fed obese mice.

## Discussion

There is strong evidence from large prospective cohort studies and meta-analyses that yogurt consumption is associated with a lower T2D risk as compared to the total dairy intake, leading to the proposal that the health benefits may be linked to the fermentation process, the lactic acid bacteria (LAB), and/or host-microbial mechanisms that are induced during yogurt consumption[2,3,27]. In this study, we report that feeding yogurt corresponding to the equivalent of two servings per day, hence replacing 7.6% of the daily energy intake, prevents insulin resistance and hepatic steatosis in diet-induced obese mice. Consistent with one of our hypotheses, we showed that the beneficial impact of yogurt on insulin sensitivity is partly driven by changes in gut microbiota composition and function (HDCA), as validated by

fecal material transplantation in GF mice. In addition, we discovered that three BCHA, derived from LAB-driven metabolism of BCAA during milk fermentation, are abundantly found in yogurt and other fermented milk products but not detectable in milk. BCHA plasma levels were positively associated with healthier metabolic parameters in obese mice and interestingly, these BCAA hydroxyl metabolites are also detected in metabolic tissues, even in non-yogurt fed mice. Remarkably, both systemic and tissue levels of BCHA are reduced in obese and insulin-resistant mice and their levels were partially maintained upon yogurt treatment, indicating that BCHA are metabolically regulated and not merely reflecting yogurt intake. We further show that BCHA can directly regulate glucose metabolism in insulin target tissues, as demonstrated by their ability to blunt glucose production and stimulate glucose uptake in liver and muscle cells, respectively.

Previous human studies reported negative associations between yogurt intake and body weight gain[4,5,20,28,29] suggesting the yogurt consumption may reduce the prevalence of obesity, although this remains to be demonstrated in randomized control trials. Here using a preclinical model of obesity, we did find a slightly lower body weight (−2.9%), result which was significant upon pooling analysis from three independent studies, and which most likely resulted from the 4% lower energy intake that was observed in yogurt-treated mice. However, it is unlikely that this minor effect on body weight can explain the better insulin sensitivity and hepatic health of yogurt-fed obese mice, especially considering that the slightly lower body weight was linked to a lower subcutaneous inguinal fat in these animals, with no changes in the most metabolically harmful visceral fat depots.

Few studies have tested the potential effect of yogurt intake on metabolic phenotypes in animal models. Lasker et al. previously reported an improvement of glucose handling following a glucose bolus in yogurt-fed obese rats, while Johnson et al. found no difference between groups of mice fed a control diet or a diet added with yogurt during an insulin tolerance test[30,31]. However, these studies were less powered, did not address the potential sites of improved glucose disposal and/or did not explore the underlying mechanisms into play. Here we observed a significant difference of fasting glucose, fasting insulin, HOMA-IR index, and the insulin response during a glucose tolerance test (GSIS) providing several indices of greater glucose homeostasis and insulin sensitivity in yogurt-treated H-fed obese mice. We have directly established the insulin-sensitizing effect of yogurt treatment by performing clamp studies, which not only confirmed the higher insulin sensitivity at the whole-body level but further indicated that both the liver and skeletal muscle contributed to

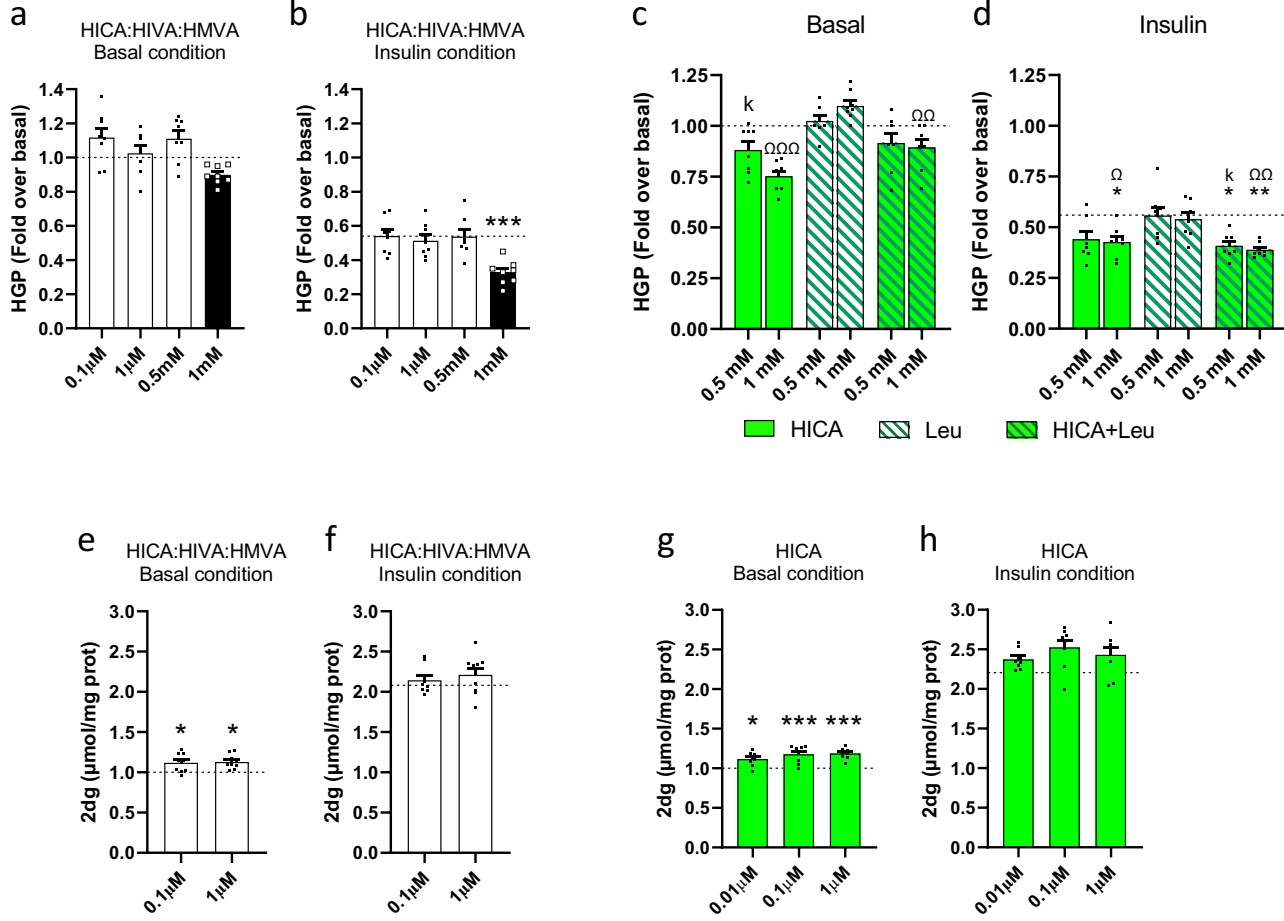

**Fig. 5 BCHA are cell-autonomous modulators of liver and muscle glucose metabolism. a** HGP of FAO cells treated with 0.1 μM, 1 μM, 0.5 mM and 1 mM mixture of HICA: HIVA: HMVA (white) at molar ratio 1: 1: 0.5 in basal condition. **b** HGP of FAO cells treated with 0.1 μM, 1 μM, 0.5 mM and 1 mM mixture of HICA: HIVA: HMVA at molar ratio 1: 1: 0.5 in insulin-treated condition. **c** HGP of FAO cells treated with 0.5 and 1 mM of HICA (lawn green), leucine (white, hatched green), or a combination of both (lawn green, hatched green) in basal condition. **d** HGP of FAO cells treated with 0.5 and 1 mM of HICA, leucine, or a combination of both in insulin-treated conditions. **e** 2dg uptake of L6 cells treated with 0.1 and 1 μM mixture of HICA: HIVA: HMVA at molar ratio 1:1:0.5 in basal condition. **f** 2dg uptake of L6 cells treated with 0.1 and 1 μM mixture of HICA: HIVA: HMVA at molar ratio 1:1:0.5 in insulin-treated condition. **g** 2dg uptake of L6 cells treated with 0.01, 0.1 or 1 μM of HICA in basal condition. **h** 2dg uptake of L6 cells treated with 0.01, 0.1, or 1 μM of HICA in insulin-treated condition. Data are expressed as mean ± SEM. Vs control: \*$p < 0.05$ versus control, \*\*\*$p < 0.001$. HICA 0.5 mM vs HIVA 1 mM: $\partial\partial p < 0.01$. Vs Leu 0.5 mM: $^K p < 0.05$, $^{KK} p < 0.01$, $^{KKK} p < 0.001$. Vs Leu 1 mM: $^\Omega p < 0.05$, $^{\Omega\Omega} p < 0.01$, $^{\Omega\Omega\Omega} p < 0.001$. The basal and insulin controls are represented by a dash line. $n = 8$–9 independent experiments. One-way ANOVA followed by Dunnett's post hoc test vs Control.

better glucose handling during insulin infusion of yogurt-fed obese mice.

Our finding that liver is a key target of yogurt action is in line with the recent finding that high consumption of yogurt (≥4 times/week) was inversely associated with the prevalence of newly diagnosed NAFLD in the general population[7]. Chen et al. further showed that when compared to milk, yogurt improved insulin sensitivity and markers of steatosis in obese women with metabolic syndrome and NAFLD[19]. Accordingly, we also observed that yogurt consumption not only preserved liver insulin sensitivity but also prevented hepatic triglyceride accumulation in H-fed mice, which was linked to lower gene expression of hepatic enzymes responsible for fatty acid metabolism. These observations were associated with slightly lower liver inflammation, NAFLD, and fibrosis scores.

The gut microbiota is increasingly recognized as a key driver of both obesity-linked T2D and hepatic diseases[32–35] and changes in gut microbiota composition were associated with yogurt intake and the increase in LAB strains were consistently observed[36,37]. Here, we report that yogurt treatment of H-fed mice induced a slight reproducible shift in the gut microbiota in all three studies with an enrichment of *Streptococcus* and a depletion of an unknown *Peptostreptococcaceae*. We further observed a few study-specific changes in bacteria from the Firmicutes phylum. In Studies 1 and 3, where we saw more robust effects of the yogurt treatment, the metabolic improvements were further associated with a lower relative abundance of bacteria from *Peptococcaceae* or *Lachnospiraceae Incertae Sedis* families, as compared to H-fed controls. Interestingly, Clarke et al. have shown a relative increase in the abundance of bacteria from *Lachnospiraceae* in diet-induced obese versus lean mice[38], whereas colonization of GF mice with *Lachnospiraceae* bacterium was reported to impair glucose tolerance and increase the weight of liver and adipose tissues[39]. Furthermore, bacteria from *Lachnospiraceae Incertae Sedis* family were reported to be increased in the gut microbiota of patients with NAFLD[40].

Bile acids (BA) are key players in the gut-liver axis, impacting both glucose homeostasis[41–46] and liver metabolism through undergoing deconjugation, dehydrogenation, and dehydroxylation in the gut altering their physico-chemical and physiological

properties[41,45,47–53]. Interestingly, we found that fecal levels of HDCA were consistently higher in H-fed mice treated with yogurt. HDCA treatment reduced hepatic triglyceride levels in mice[54] and decreased blood glucose in LDL receptor KO mice[55], while a decrease in HDCA was reported as the main BA alteration in a rat model of hypertensive NAFLD[56]. Collectively, the changes in the gut microbiota composition together with an increase in HDCA found in yogurt-treated obese mice likely contribute to the healthier metabolic phenotypes.

FMT studies in GF mice further established the causal role of the gut microbiota changes in some of the beneficial effects of yogurt treatment in obese mice. Indeed, transplantation of fecal slurries from yogurt-treated mice to GF mice promoted lower fasting insulinemia and HOMA-IR, as well as lower insulin levels during the glucose tolerance test. However, FMT studies failed to reduce liver weight and hepatic steatosis in GF mice, suggesting that the healthier liver phenotypes may not be driven by the shift in the gut microbiota. Alternatively, the microbial species responsible for the hepatic benefits of yogurt intake may have been lost during processing or the transfer of microbial communities to GF mice, such as those species involved in HDCA production. Indeed, the colonization of GF mice remains a reductionist approach, whereby only a fraction of the donor microbiota is expected to survive sample collection, fecal preparation and storage, and passage through the recipient's stomach to find a niche in the gut.

In the search for potential liver metabolites that may be associated with the health benefits of yogurt intake, we found that yogurt administration preserved hepatic levels of three hydroxyacids (HIVA, HICA, and HMVA) that are derived from BCAA catabolism. Indeed, HIVA, HICA, and HMVA are generated following the catabolism of BCAA and reduction of their cognate ketoacids to hydroxyacid intermediates, a process that is well conserved and common in yeasts, fungi, and bacteria including LAB[57]. BCAA levels did not differ between diets (3.5 vs 3.4 g/100 g of H diet and Y diet, respectively), further suggesting that the difference in BCHA in the experimental groups may come from yogurt starter activity, and/or from different microbial and/or host metabolism of BCAAs. Bioinformatics search revealed that both *L. bulgaricus* CNCM I-1519 and *S. thermophilus* CNCM I-1630 strains harbor gene with identity to hydroxyacid-dehydrogenases. Although LAB activity on yogurt BCAA catabolism explains the enrichment of BCHA in yogurt versus milk, it is also of major interest that these metabolites are readily detected in the circulation, liver and skeletal muscle of non-yogurt exposed lean mice and significantly reduced upon H-fed diet-induced obesity. To the best of our knowledge, this is the first study reporting that obesity is linked to reduced levels of all three BCHA in the plasma and metabolic tissues. The levels of these metabolites were, in addition, negatively associated with plasma glucose as well as liver triglycerides, suggesting that they are novel biomarkers of metabolic and liver health. Importantly, we also found that yogurt administration of H-fed obese mice partially preserved the levels of BCHA in both the plasma and tissues of these animals. Since the dose of yogurt used in these studies corresponds to two servings of yogurt per day in humans, it can be assumed that the resulting BCHA levels are metabolically relevant. Interestingly, BCAA catabolism by LAB yields flavor compounds, carboxylic acids and aldehydes, and also BCHA that are devoid of aroma[58]. In cheese, limiting hydroxyacid dehydrogenase activity is, however, preferred to enhance cheese aroma and this could underlie lower BCHA levels and contribute to the epidemiological differences in T2D risk reduction of fermented dairy[12,17]. Further and importantly, we showed that BCHA can directly modulate glucose metabolism in liver and muscle cells and that even very low concentrations of

these metabolites, corresponding to their levels measured in the circulation, are able to increase glucose uptake by myocytes. Although higher levels of BCHA were needed to blunt glucose production by hepatocytes it is speculated that higher levels of these BCAA metabolites are released in the portal circulation to impact on liver metabolism. These data thus provide evidence that BCHA are cell-autonomous modulators of glucose metabolism and may be involved in the beneficial metabolic effects of yogurt treatment through their action on both liver and muscle tissues.

Future studies will be needed to determine the mechanisms underlying our findings of reduced BCHA levels in obesity. In contrast to BCAA metabolism by LAB and other microorganisms, very little is known about BCHA production or metabolism by commensal gut microbes and host mammalian tissues. It will be important to determine whether some of the bacterial taxa found to be altered by obesity and yogurt treatment express the enzymes required to metabolize BCAA to BCHA and whether this could contribute to the yogurt effect. It is also possible that alterations in liver and muscle BCAA metabolism in obesity contribute to the lower production of their BCHA congeners. Indeed, maple syrup urine disease (MSUD) caused by deficient activity of the branched-chain alpha-ketoacid dehydrogenase complex (BCKDH) is characterized by increased plasma concentrations of BCAA but also of their keto-intermediates (2-ketoisovaleric acid, 2-ketoisocaproic, and 2-keto-3-methylvaleric acid) and BCHA[59]. Although MSUD is an extreme example of mammalian BCHA metabolism, it shows that the enzymatic machinery to catabolize BCAA to BCHA exists in humans. In line with this, the lactate dehydrogenase LDHA was shown to produce the L- form of HIVA in vitro[60] and elevated D-forms of HIVA and HICA were observed in patients with missense variants in LDHD gene[61]. Considering that altered BCAA metabolism is a pathognomonic feature of insulin resistance and T2D, the BCAA to BCHA conversion could possibly be harnessed for therapeutic applications[23,62–68]. Further work is necessary to unravel the host-microbial mechanisms driving the BCAA metabolism to BCHA and whether targeting the enzymatic machinery involved in the production of these hydroxyl BCAA congeners could improve insulin sensitivity and liver diseases in obesity. BCAA catabolism to BCHA is well documented in bacteria but poorly characterized in mammals and thus represents a major gap in our understanding of BCHA role and regulation by mammalian orthologs of these key BCAA catabolic enzymes. However, our findings show that yogurt intake over only three months can partially prevent BCHA reduction in obese animals, suggesting that nutritional interventions with yogurts, and possibly other fermented dairy products, should be explored in clinical trials to determine whether they can supply exogenous BCHA and compensate for their impaired endogenous production, and thus alleviate insulin resistance and NAFLD in obesity.

## Methods
### Lyophilized yogurt product and diet preparation
*Lyophilized yogurt product.* The yogurt (1.9% fat, 4.7% protein, 6.6% carbohydrate (g/100 g of product), *L. bulgaricus* CNCM I-1519, *S. thermophilus* CNCM I-1630) was used due to its representative macronutrient composition (un-supplemented with calcium, vitamin D, starch, no fruits or flavors, protocooperation of the two strains) and freeze-dried. For all studies, the yogurt was produced at the Danone Nutricia Research site (Study 1 and 2 multiple productions, Study 3 single production) following bovine milk, cream, milk powder hydration, and pasteurization at 95 °C for 5 min. The yogurt was fermented using the industrial (Hansen) cultures of *L. bulgaricus* CNCM I-1519 and *S. thermophilus* CNCM I-1630 (ratio 5:95) at 39 °C for 9 h with breaking at pH = 4.65. Following smoothing at 3000 RPM, the yogurt was stored at 4 °C for 3 days before lyophilization (Studies 1 and 2). Briefly, yogurt freezing (−42 °C for 17 h) was followed by two steps of sublimation (−20 °C for 30 h30 and −10 °C for 17 h) and bound-linked water was

**Table 2 Nutritional composition of Y lyophilized product.**

| Nutritional composition (g per 100 g of product) | Study 1 | Study 2 | Study 3 |
|---|---|---|---|
| Fat | 12.3 | 13.1 | 13.2 |
| Carbohydrate | 45.0 | 43.8 | 43.5 |
| Protein | 31.3 | 31.2 | 31.7 |
| Val. Cal. (Kcal) | 415.7 | 417.9 | 419.6 |
| Fructose | <0.2 | <0.2 | <0.2 |
| Glucose | 0.4 | <0.2 | <0.2 |
| Lactose | 35.5 | 33.5 | 33.6 |
| Maltose | <0.2 | <0.2 | <0.2 |
| Sucrose | <0.2 | <0.2 | <0.2 |
| Galactose | 4.4 | 5.4 | 4.6 |
| Calcium | 1.2 | 1.2 | 1.1 |
| Sodium | 0.4 | 0.4 | 0.3 |
| **Microbiological profile (log CFU per 100 g)** | **Study 1** | **Study 2** | **Study 3** |
| *L. bulgaricus* CNCM I-1519 | 7.1 | 7.1 | 6.9 |
| *S. thermophilus* CNCM I-1630 | 9.1 | 9.1 | 9.4 |

removed by desorption (20 °C for 24 h), which resulted in the lyophilized yogurt product (Studies 1 and 2). In Study 3, one batch of 43 kg of yogurt was conditioned, shipped, and stored at 4 °C for 2 days before lyophilized at Lyofal, France with a Usifroid SMH 150 lyophilizer. The stability of yogurt and lyophilized product was monitored throughout protocol: until day 50 (Study 1), day 80 day (Study 2) and day 230 (Study 3). The *L. bulgaricus* CNCM I-1519 count was 7.1 ± 0.1 log CFU/g on day 50 (Study 1), 7.0 ± 0.2 log CFU/g on day 80 (Study 2) and 6.9 log CFU/g on day 230 (Study 3). The *S. thermophilus* CNCM I-1630 count was 9.2 ± 0.2 log CFU/g on day 50 (Study 1), 8.8 ± 0.2 log CFU/g on day 80 (Study 2) and 9.5 log CFU/g on day 230 (Study 3). In Study 1, lyophilized yogurt was consumed by mice between day 13 and 42 post-production and nine productions were needed to cover the study. In Study 2, lyophilized yogurt was consumed by mice between day 29 and 80 post-production and five productions were needed. A single batch of lyophilized yogurt was consumed by mice through the Study 3. The lyophilized yogurt was stored at 4 °C and the Y diet was formulated daily using the pre-conditioned sealed bags. Nutritional and microbiological profiles (Enterobacteria < 100 CFU/g, yeast, and mold <10 CFU/g; NF ISO 215 28-2; ISO 6611) of different lyophilized products were analyzed and checked whether well concentrated versus the initial yogurt. Nutritional composition (ISHA and ITERG, France) and microbiological profile (plate count method ISO 7889:2003 IDF 127, accuracy of ±0.5, mean of productions over time) of lyophilized product is available in Table 2. Some other dairy products were also used compared to the studied yogurt product and their nutritional profile and species used for fermentation are detailed in Supplementary Fig. 6.

*Dietary treatments.* Mice were fed with either a low-fat low-sucrose (C) or a high-fat high-sucrose diet containing a protein mixture representative of the United States diet (H)[69]. A third group was fed high-fat high-sucrose diet that contained a protein mixture representative of the US diet in which lyophilized yogurt was incorporated (Y). The protein sources for mixture incorporation in diets consisted of cooked meats (chicken/pork/beef, Happy Yak, Canada), soy (Teklad Envigo, USA), white egg (Teklad Envigo, USA) and dairy (casein, MP biomedicals, Canada). All the protein sources were provided in a lyophilized form, and details of their process are available on the respective websites of the providers, with the exception of beef, pork, and chicken, which were lyophilized lean cuts of meat without any additional processing. The composition of each protein source has been analyzed (EnvironeX) and the resulting protein mixture consisted of 82% protein, 0.5% carbohydrate, and 13% fat. Diets were then matched for protein, fat, carbohydrate, and saturated: polyunsaturated fatty acid ratio (SAT: PUFA). For more information, see Choi et al.[22].

The three animal groups were fed with either a C (3.7 kcal/g), a H (4.8 kcal/g) or a Y (4.8 kcal/g) diet. The Y diet was formulated so that 8.6% of control H diet was substituted by lyophilized yogurt and the caloric content, the macronutrients, and the saturated to polyunsaturated fatty acids ratio remain adjusted (Table 1). Branched-chain amino acid content (BCAA) was analyzed and equivalent to 3.5 g/100 g of diet for H diet (1.6 g/100 g leucine, 0.9 g/100 g isoleucine, 1.1 g/100 g valine) and 3.4 g/100 g of diet for Y diet (1.5 g/100 g leucine, 0.9 g/100 g isoleucine, 1.1 g/100 g valine). The lyophilized yogurt replacement represented ~7.6% of daily energy intake, which corresponded to ~2 servings (225–250 g) of yogurt per day in humans. The lyophilized yogurt was manually incorporated into the H diet every day and all the diets (C, H, Y) were changed daily prior to the onset of the dark cycle, in a way that fresh food was presented to the mice just before the light switches off. The viability and count stability of both bacteria in Y diet were ensured over 20 h at 22 °C using plate count method.

*Germ-free study.* One week before fecal material transplantation and for all the 9-week duration of the protocol, GF mice were fed with H diet previously supplemented with 50% more vitamins and irradiated twice (29.1–41.6 kGy, Nordion, Laval, QC).

**Animals.** All procedures were previously approved by the Laval University Animal Ethics Committee (Québec, Canada), referred to the 2016-004-21 protocol and followed the internal *Guide for the care and use of laboratory animals*.

*Conventional animals.* Six-week-old male C57Bl/6 mice (Jackson, USA) were single-housed in the animal facility of the Quebec Heart and Lung Institute (Québec, Canada) in a controlled environment (12 h daylight cycles, lights off at 6:00 PM, 22 °C) with food and water ad libitum for all studies. After two weeks of acclimatization on C diet, baseline metabolic measurements were assessed. Animals were then allocated to treatment groups matching for body weight, and dietary challenged for 12, 15, and 12 weeks respectively in Study 1 ($n = 18$–24 per group), 2 ($n = 14$–24 per group) and 3 ($n = 36$ per group). Food intake and body weight were measured daily and weekly, respectively. Body composition was analyzed via quantitative NMR spectroscopy (Bruker minispec LF90) before and at 12 and 15 weeks of diet challenge in Studies 1 and 2, respectively.

*Germ-free animals.* Male Swiss Webster mice (Tac:SW genetic background), were bred and housed in the GF animal facility of the IUCPQ Research Center with free access to water and irradiated chow diet. At eight-week old, animals were fed with H diet. At nine-week old, mice were allocated to treatment groups taking care of litter and dam and matching for their body weight. Four groups of 11–12 animals were formed, each constituted of three isocages of $n = 3$–4 mice. For FMT, feces were collected at week 12 in Study 1 from mice fed with H or Y diets. Three pools of feces for each treatment group were formed and resuspended with sterile saline (1:10 dilution). The resultant solution was homogenized, spin-down and 200 μL were gavaged to the receivers housed in the same cage. The remaining pellet of the solution was spread on mice nests. Mice were gavaged only once with fecal slurries as it was previously shown to be sufficient to establish microbiota in isocage system conditions[70]. Mice were maintained in the GF facility and fed the H diet for nine weeks. A separated FMT study was also performed with another cohort of GF mice that were kept on a chow diet after fecal microbiota transfer from H1 and Y1-fed donors. Body weight was recorded weekly and food intake was changed and recorded twice a week.

Mice were euthanized under isoflurane anesthesia by cardiac puncture. In Studies 1 and 2, mice underwent a blood collection after 4 and 6 h fasting, respectively, before being euthanized. In Study 3, mice were euthanized right after the clamp experiment. In the GF Study, mice were euthanized after undergoing an oGTT (4 h fasting, 2 h of test). Blood was drawn and immediately centrifuged, and plasma was collected and stored at −80 °C. Relevant tissues were harvested, weighed, snap-frozen in liquid nitrogen, and stored at −80 °C for further analysis.

**Glucose homeostasis.** All tests requiring blood samples were performed via saphenous vein puncture after 4 h fasting for Study 1, 3, and GF, or 6 h fasting for Study 2. For fasting glucose and insulin measurements, blood samples were withdrawn before and at the end of the treatment, except for the GF experiment. An oGTT was performed at week 11 in Studies 1 and 2, and at week 9 for the GF study. Glycemia was recorded before (0 min) and after (15, 30, 60, 90, 120 min)

after a glucose challenge administrated by gavage (1 g/kg body weight). Insulin response was measured in plasma of half of the animals (time 0, 15, 30, 60, and 120 min) in Study 1, and all the animals in Studies 2, 3, and GF. The C-peptide was measured in animals for which remaining plasma was available in Study 1.

**Hyperinsulinemic-euglycemic clamps (HIEC)**. After 12 weeks of dietary intervention in Study 3, HIEC were performed in conscious unrestrained mice as previously described[71]. The left common carotid artery and the right jugular vein were catheterized 3 days before the experiment for blood sampling and infusions, respectively. A blood sample was collected for measuring basal glucose and insulin levels 90 min prior to the clamp start. Then, the $[3-^3H]$-glucose (Perkin Elmer, Boston, MA) was prime-infused at 0.33 μCi/min during 90 min for equilibration ($-90$ to 0 min). The insulin clamp started upon primed infusion of insulin (2.5 mU/min/kg; Humulin R; Eli Lilly Canada Inc., Scarborough, ON) and the $[3-^3H]$-glucose infusion was increased to 0.67 μCi/min ($t = 0$ to 120 min). Euglycemia was maintained at $7 \pm 0.5$ mM level by infusing 20% dextrose as needed. Blood samples (5 μL) were collected every 10 min from 0 to 120 min to measure blood glucose and adjust dextrose infusion in consequence. Blood samples (30–50 μL) were collected every 10 min from $-30$ min to 0 min and from 90 to 120 min to calculate basal and clamped Ra and Rd, respectively. Clamp insulin levels were determined from samples obtained at 120 min. A saline-washed erythrocytes solution from mice donors was infused (0.1 mL/hr) throughout the HIEC to prevent a decrease in the hematocrit.

**Analytical methods**. Blood glucose was recorded using a One Touch glucometer UltraMini® and values higher than the upper limit of glucometer were determined by an Amplex® red glucose/glucose oxidase assay kit. Plasma insulin and C-peptide were measured by Mouse ELISA kits (ALPCO Ultrasensitive Insulin and Crystal Chem Inc. C-peptide). Homeostasis model assessment of insulin resistance was calculated using the formula: fasting insulin (microU/mL) × fasting glucose (mmol/L)/22.5. Standard colorimetric kits were used for the measurement of plasma triglycerides (Infinity™), free fatty acids (WAKO HR series NEFA-HR (2)), and total cholesterol (Randox Life Science). ELISA kits were employed for adiponectin (Millipore Sigma), Lipopolysaccharide (LPS) endotoxin (MyBioSource), and LPS Binding Protein (Hycult Biotech) determinations. Corticosterone was measured by an enzyme immunoassay kit (Arbor Assay). Leptin and PYY plasma levels were assessed by a Milliplex Map Mouse Metabolic Magnetic Bead Panel while chemokines and cytokines were assessed in plasma and in tissue lysates (in PBS containing 0.01% protease inhibitor) using a Bio-plex Pro™ Mouse cytokine standard 23-plex (Bio-Rad) in Study 1. For Study 2, cytokines were analyzed by an ELISA. Liver triglycerides and cholesterol were extracted by an adapted Folch method and dosed with the same kits used for plasma analysis.

**Tissues mRNA gene expression by real-time qPCR**. Liver and ileum gene expression was analyzed by RT-qPCR. Samples were homogenized in TRIzol (Invitrogen) and RNA was extracted using the GeneJET RNA Purification Kit (Thermo Scientific). The cDNA was synthetized (High-Capacity cDNA Reverse Transcription Kit, Applied Biosystems) and subjected to qPCR analysis (CFX384 Touch, Bio-Rad Laboratories) using TaqMan® gene expression assays (listed in Supplementary Data 5 and 6) and TaqMan® Fast Advanced Master Mix (Thermo Scientific), according to the manufacturer's instructions. mRNA expression of each gene was normalized to three references genes (*Gapdh*, *Mrpl19*, and *Hprt* for liver, *Actb*, *Gapdh*, and *Hprt* for ileum) using CFX Manager 3.1 Software (Bio-Rad Laboratories).

**Hepatic steatosis/fibrosis staining and scoring**. Formalin-fixed paraffin-embedded liver samples were cut (4 μm) and stained with hematoxylin and eosin (H&E) on an automatized platform for histologic assessment. Liver sections were blindly evaluated by a pathologist medical trainee and revised by a pathologist at the hospital IUCPQ center using the NAFLD scoring system proposed by Liang et al.[72]. The percentage of hepatocytes harboring either macro, microvesicular steatosis, or hypertrophy (increase of more than 1.5 the size of a normal hepatocyte) was scored as steatosis. Inflammation was evaluated as the number of inflammatory foci (cluster of at least five inflammatory cells) per field at ×10 magnification (3.5 mm²) in at least five fields. The presence of steatosis in more than 5% of hepatocytes was classified as NAFLD and the association of both steatosis and inflammation in more than 0.5 cluster per field was classified as NASH. Other paraffin-embedded sections (6-μm thick) were deparaffinized through toluene and a graded alcohol series to assess liver fibrosis. Sections were stained for 8 min with Weiger Hamatoxyline and rinsed for 5 min in water. Subsequently, the sections were stained for 60 min with Sirius red (0.1% of Sirius red in saturated aqueous picric acid) for collagen bundle staining. Sections were then rinsed with acetic acid 0.05% and mounted for observation under polarized light microscopy (Zeiss AXIOPLAN; Oberkochen, Germany). Fibrosis was scored according to the definition used by Kleiner et al.[73]. A score of 1 was given when fibrosis was found in perisinusoidal or periportal regions, 2 when fibrosis was present in both, 3 with bridging fibrosis, and 4 with cirrhosis.

**Targeted lipids and untargeted metabolites profiling**. The targeted lipidomics (CLP platform) and untargeted metabolomics (HD4 platform) analysis were performed by Metabolon (Morrisville, North Carolina, USA)[74]. Lyophilized yogurt and lyophilized milk ($n = 5$ of each and representing different productions) and liver biopsy from 20 mice (four from group C, eight from group H and eight from Group Y selected as representatives of each respective group hepatic triglycerides) from Studies 1 and 2 were used to generate profiles. Flash-frozen samples (100 mg of liver biopsy, 300 mg of lyophilized yogurt and milk) in liquid nitrogen were stored at $-80$ °C and shipped to Metabolon on dry ice.

*Sample preparation HD4*. Detailed description of sample preparation, gradients, standards, and instrumental parameters have been published[75]. Recovery standards were added prior to the first step in the extraction process for quality control purposes. Briefly, liver samples were soaked overnight in MeOH in a 25:1 V:W ratio. Milk powder samples were reconstituted with a 50:1 V:W ratio of water to render liquid samples. For liquid samples, including QC samples, 100 μL of material was extracted with methanol under vigorous shaking for 2 min (Glen Mills GenoGrinder 2000) to precipitate protein and dissociate small molecules bound to protein or trapped in the precipitated protein matrix, followed by centrifugation to recover chemically diverse metabolites. The resulting extracts were divided into multiple fractions: two for analysis by two separate reverse-phase ultrahigh-performance liquid chromatography-tandem mass spectrometry (RP)/UPLC-MS/MS methods using positive ion mode electrospray ionization (ESI), one for analysis by RP/UPLC-MS/MS using negative ion mode ESI, one for analysis by HILIC/UPLC-MS/MS using negative ion mode ESI, and others reserved for backup. Samples were placed briefly on a TurboVap® (Zymark) to remove the organic solvent. The sample extracts were stored overnight under nitrogen before preparation for analysis.

*Sample preparation CLP*. Liver samples were soaked overnight in DCM:MeOH in a 30:1 V:W ratio. 240 μL of the resultant extract were further processed with a modified Bligh-Dyer extraction[76,77] using methanol/water/dichloromethane in the presence of deuterated internal standards (Supplementary Data 7). Milk powder samples were reconstituted with a 50:1 V:W ratio in water and 50 μl was extracted using the same modified Bligh-Dyer extraction. The extracts were dried under nitrogen prior to instrumental analysis.

*HD4 UPLC-MS/MS*. Detailed description of gradients, standards, and instrumental parameters have been published[1]. Briefly, methods utilized a Waters ACQUITY UPLC, and a Thermo Scientific Q-Exactive high resolution/accurate mass spectrometer interfaced with a heated electrospray ionization (HESI-II) source and Orbitrap mass analyzer operated at 35,000 mass resolution. The dried sample extract was reconstituted in solvents compatible to each of the four methods. Each reconstitution solvent contained a series of standards at fixed concentrations to ensure injection and chromatographic consistency. One aliquot was analyzed using acidic positive ion conditions, chromatographically optimized for more hydrophilic compounds. In this method, the extract was gradient-eluted from a C18 column (Waters UPLC BEH C18-2.1 × 100 mm, 1.7 μm) using water and methanol, containing 0.05% perfluoropentanoic acid (PFPA) and 0.1% formic acid. A second aliquot was also analyzed using acidic positive ion conditions but was chromatographically optimized for more hydrophobic compounds. In this method, the extract was gradient eluted from the C18 column using methanol, acetonitrile, water, 0.05% PFPA, and 0.01% formic acid, and was operated at an overall higher organic content. A third aliquot was analyzed using basic negative ion optimized conditions using a separate dedicated C18 column. The basic extracts were gradient-eluted from the column using methanol and water, however with 6.5 mM Ammonium Bicarbonate at pH 8. The fourth aliquot was analyzed via negative ionization following elution from a HILIC column (Waters UPLC BEH Amide 2.1 × 150 mm, 1.7 μm) using a gradient consisting of water and acetonitrile with 10 mM ammonium formate, pH 10.8. The MS analysis alternated between MS and data-dependent $MS^n$ scans using dynamic exclusion. The scan range varied slightly between methods, but covered ~70–1000 $m/z$.

*CLP FIA-DMS-MRM*. The CLP methodology is a flow injection analysis (FIA), differential mobility separation (DMS), and multiple reaction monitoring (MRM) MS analysis. The CLP assay targets 1100+ lipid species covering 14 different lipid classes (Supplementary Data 8) and 29 fatty acid constituents. The CLP extracts were reconstituted solvent is 1:1 DCM:MeOH with 10 mM NH4Ac were transferred to glass-lined plates for infusion-MS analysis, performed on a Shimadzu LC with nano PEEK tubing and the Sciex SelexIon-5500 QTRAP. The CLP samples were analyzed via both positive and negative mode electrospray as appropriate for each lipid class (Supplementary Data 8). The lipid extract was continuously infused into the mass spectrometer without the use of any chromatographic column at 8 μL/min. A uniform sample was analyzed throughout the infusion (~6 min), allowing 17 replicate measurements per lipid species for more robust and reproducible results. After ionization in the source of the mass spectrometer, lipids were introduced into the DMS cell. The DMS cell acted as a lipid filter or gate that permitted a specific lipid class to pass into the mass spectrometer at a time, while the other lipid classes were filtered out. The DMS cell cycled through the different lipid classes over the course of a single sample infusion, sequentially passing each

lipid class into the mass spectrometer for MRM analysis. When applicable the DMS functionality was utilized to allow unambiguous lipid identification. In the MRM analysis, the first quadrupole (Q1) filtered based on the $m/z$ of the intact lipid species, while the third quadrupole (Q3) filtered based on the $m/z$ of a characteristic fragment of that same lipid species, such as one of the fatty acid side chains (Supplementary Data 8).

*HD4 data extraction and compound identification.* Raw data were extracted, peak-identified, and quality control processed using Metabolon's hardware and software. These systems were built on a web-service platform utilizing Microsoft's.NET technologies and compounds were identified by comparison to library entries of purified standards or recurrent unknown entities. Metabolon maintains a library based on authenticated standards that contain the retention time/index (RI), mass to charge ratio ($m/z$), and chromatographic data (including MS/MS spectral data) on all molecules present in the library. Furthermore, biochemical identifications are based on three criteria: retention index within a narrow RI window of the proposed identification, accurate mass match to the library ±10 ppm, and the MS/MS forward and reverse scores (MSI level 1, unless designated with an *). MS/MS scores are based on a comparison of the ions present in the experimental spectrum to ions present in the library entry spectrum. While there may be similarities between these molecules based on one of these factors, the use of all three data points can be utilized to distinguish and differentiate biochemicals. More than 4500 commercially available purified standard compounds have been acquired and registered into LIMS for analysis on all platforms for the determination of their analytical characteristics. Additional mass spectral entries have been created for structurally unnamed biochemicals, which have been identified by virtue of their recurrent nature (both chromatographic and mass spectral).

*Compound quantification.* HD4 method compounds were quantified as area-under-the-curve detector ion counts and raw data were scaled for purposes of data visualization. For CLP compounds, individual lipid species were quantified by taking the ratio of the signal intensity of each target compound to that of its assigned internal standard, then multiplying by the concentration of internal standard added to the sample. CLP used a range of internal lipid standards (Supplementary Data 7) which for most classes covered both carbon length and number of double bonds. Experimental lipids were matched to specific standards based on the combination of carbon length and number of double bonds. Lipid class concentrations were calculated from the sum of all molecular species within a class, and fatty acid compositions were determined by calculating the proportion of each class comprised by individual fatty acids.

*Data quality control.* Several types of quality control measures were used in concert with the experimental samples. These included: (1) technical replicate samples generated by combining a small portion of each experimental sample spaced evenly among experimental samples; (2) extracted water samples (process blanks) and solvent blanks; and (3) a cocktail of quality control standards[75] spiked into every analyzed sample. Instrument variability was determined by calculating the median relative standard deviation (RSD) for the standards that were added to each sample prior to injection into the mass spectrometers. Overall process variability was determined by calculating the median RSD for all endogenous metabolites (i.e., non-instrument standards) present in each of the technical replicate samples. Experimental samples were randomized across the platform run, with quality control samples spaced evenly among the injections. The median RSD was for internal standard 4% liver and 3% milk product, for endogenous metabolites 7% liver and 7% milk product and finally, for complex lipid 5% liver and 4% milk product. In addition, a variety of procedures were performed to ensure that a high-quality data set was made available for statistical analysis and data interpretation. Each compound was reviewed for accurate and consistent peak identification, alignment, and quantitation. Peaks/compound calls were removed if the represented system artifacts, mis-assignments, redundancy, and/or background noise.

**Statistical analyses** (significance tests and classification analysis) were performed by Metabolon. Missing values were imputed with the observed minimum for each compound. Statistical analyses were performed on log-transformed data using two-way ANOVA with Study, Group (diet or product) and their interaction, separately for metabolomic HD4 and lipidomic CLP datasets. Multiple testing correction was done using FDR (q-value) comparison-wise. Analyses were performed using ArrayStudio.

**NMR targeted analysis.** The sample preparation and the NMR analysis were performed by Bioaster (Lyon, France). The lyophilized milk (100 mg, −80 °C storage), lyophilized yogurt (100 mg, Study 1, 2 and 3, −80 °C storage), milk (20 mL, $n = 1$ of 0% fat milk, $n = 1$ of 2% fat milk, $n = 3$ of whole fat milk), non-lyophilized yogurt (−80 °C storage), yogurt 1 to 3, fermented milk 1 and 2 and skyr (1 ml each) were shipped to Bioaster ($n = 5$ per each sample and storage at 4 °C unless stated otherwise). The chemical standards were HMVA, HIVA, and HICA (Sigma Aldrich Ref. 80529, Ref. 219835, and Ref. 219827, respectively).

*Sample preparation of lyophilized products.* Hundred milligrams of lyophilized products were solubilized in 1 mL of MiliQ water, vortexed shortly, sonicated for

10 min, and then centrifuged 10 min (10 °C, 10,000 × $g$). Recovered 200 µL of clean supernatants were additionally ultra-filtered using 10 kDa cut-off filtration tubes for protein depletion during 50 min at 10 °C and 10,000 × $g$ centrifugation speed.

The 135 µL of ultra-filtrates were mixed with 45 µL of 2 mM sodium 2,2-dymethyl-2-silapentane-5-sulfonate (DSS, Eurisotop) internal standard solution containing 1 M phosphate buffer (pH = 7.4) and $D_2O$ for the signal lock in 60:40 v/v ratio. The final concentration of the DSS was 0,5 mM. The obtained solutions were centrifuged for 5 min at 10 °C and 10,000 × $g$ speed and 155 µl of the supernatants were transferred in 3 mm SampleJet NMR tubes. In addition to samples analysis, blank samples were also prepared using the same sample preparation protocol, but for which the milk-derived product was replaced by MiliQ water.

*Sample preparation of fermented products.* Fermented milk products were shacked 10 min using tube rotator (Stuart) with 15 mL Falcon adapter. In addition, before sampling, all tubes were vortexed 15 s and then 1 mL of samples were transferred in 1,5 mL Eppendorf tubes. The tubes were centrifuged for 10 min at 10 °C and 10,000 × $g$ speed in order to obtain a clean supernatant. Next, 200 µL of the supernatants were ultra-filtred with 10 kDa cut-off filtration tubes and then the filtrates were processed in the same way as described above for lyophilized products.

*Sample preparation of milk.* Liquid milk (1 mL) was transferred to 1.5 mL Eppendorf tube and centrifuged 10 min at 10 °C and 10,000 × $g$. The 200 µL of supernatants were ultra-filtered by centrifugation using 10 kDa cut-off filtration tubes for protein depletion during 50 min (10 °C, 10,000 × $g$). The resulting ultra-filtrated milk samples were processed as mentioned above for lyophilized products.

*Sample preparation for accuracy and precision evaluation.* To 330 µL of milk, fermented products or suspended lyophilized products in MiliQ water at a rate of 100 mg to 1 mL was spiked with 20 µL stock solutions containing hydroxyl metabolites at low, medium and high concentrations and the background samples was spiked with MiliQ water. All spiked samples prepared in 1.5 mL Eppendorf tubes ($n = 3$) were vortexed 15 s and mixed on the Thermoshaker (Eppendorf) during 10 min at 10 °C and 1000 rpm.

*NMR analysis and processing.* The sample analysis was performed using Ascend 600 MHz NMR spectrometer from Bruker Biospin, (Lyon, France) equipped with cryogenically cooled 5 mm QCI (1H/13 C/15 N/31 P) probe head. All samples were stored at 4–6 °C in SampleJet autosampler before analysis. For each sample, a one-dimensional proton acquisition was performed using noesygppr1d sequence, which contains a pre-saturation block for water attenuation during relaxation delay. The spectral width of proton acquisition was 12 ppm using 32 k data points during 2.3 s acquisition time and 4 s relaxation delay. The mixing time delay was 80 ms. All spectra contained 128 scans for about 14 min total acquisition time. The free induction decay raw data were processed by Fourier transformation using 0.3 Hz exponential apodization function within 32 k data points.

*Metabolite quantification.* The resulting spectra were phased, and baseline corrected. The spectra alignment was done according to internal standard DSS chemical shift. Finally, the metabolites quantification was performed using Chenomx NMR suite 8.3. The metabolites library database was developed with the help of Compound Builder module using experimental proton spectra of a known concentration. The limit of detection was 1.2 µM for HICA (0.915 ppm, dd, CH3), 1.2 µM for HIVA (0.825 ppm, d, CH3), and 2.5 µM for HMVA (0.78 ppm, d, CH3). The data are reported as a mean and standards deviation.

**Targeted LC-MS/MS analysis.** The samples preparation and the targeted LC-MS/MS analysis were performed by Bioaster (Lyon, France). The plasma, liver, and gastrocnemius tissues sample (4–5 from group C; 8 from group H; 8–9 from Group Y) were used to measure HICA, HIVA, and HMVA quantity. Flash-frozen samples (>30 µL of plasma, >100 mg of liver biopsy, >200 mg of gastrocnemius biopsy) in liquid nitrogen were stored at −80 °C and shipped to Biaoster on dry ice. The chemical standards were HMVA, HIVA and HICA (as above) and the internal standard was 2-hydroxy-2-methylbutyric acid 98% (Sigma Aldrich Ref. H40009).

*Sample preparation.* The plasma samples (30 µL) were thawed and mixed with ice cold methanol (105 µl) for proteins precipitation during 10 min at 1000 rpm and 4 °C. Next, for each sample 100 µl of supernatant was recovered by centrifugation and transferred in a clean tube for evaporation. The dry residue was suspended in 100 µL methanol:water solution (85:15 v/v) for further LC-MS/MS analysis.

The liver and gastrocnemius tissues were weighted and homogenized using the Precellys system and a biphasic (Bligh-Dyer) approach was used for metabolites extraction. Thus, 400 µl of methanol, 200 µl of chloroform, and 160 µl of water were added to the homogenization tubes and the tissues were homogenized using the following parameter: 10,000 rpm, 3 × 40 s and pause 30 s, 4 °C. After 15 min of incubation at 10 °C and 1000 rpm using a Thermoshaker (Eppendorf), the mixes were centrifuged, and the supernatant was transferred in a clean 5 mL Eppendorf tube. The remaining tissues were extracted again with fresh solvents, following the

same protocol. The new extract was combined with the first one, for a total of 1,1 mL. The phase separation was carried out by the addition of 200 μL of water and 200 μl of chloroform to extracts and the top fractions were collected and dried under nitrogen stream. The residues were suspended in 85 μl of water and 15 μl of methanol for further LC-MS/MS analysis.

*LC-MS/MS.* The metabolites quantitation was based on the protocol developed by Ehling and Reddy[78,79]. Thus, the analysis was performed using an HPLC (Ultimate 3000 Thermo) coupled to a Triple Quadrupole Mass spectrometer (Quantum Ultra Thermo). The chromatographic separation of the metabolites was carried out on a Waters Aquity BEH C18 1.7 μm 2.1 × 100 mm column thermostated at 40 °C and gradient eluted with water containing 0.1% formic acid as mobile phase A and the methanol with 0,1% formic acid as mobile phase B. Thus, the solvents flow rate was 300 μL/min and the gradients were as follows: it starts with 10% B; then 0–2 min the B increased to 15% and remain constant 8 min; from 10 to 10.5 min the B increased to 100% and remain constant for 2 min; from 12.5 to 13.5 the mobile phase B was returned back to 10% and maintained for equilibration 3 min. The sample injection volume was 10uL. The MS analysis was performed in negative electrospray ionization mode (ESI). Source settings were as follows: the spray voltage 3 kV; the vaporizer temperature 100 °C the Sheath and auxiliary gas pressure were 15 and 10 respectively; the capillary temperature 350 °C; and the tube lens and skimmer offset were 46 and 10 V respectively.

*BCHA detection and quantification.* The standards for each metabolite were detected in MRM mode and well separated, with distinct retention times: HIVA (117 ® 71 *m/z*, 13 V (CE)), 3.6 min; HMVA metabolite was detected under two diasteriomeric forms HMVA1 (131® 85 *m/z*, 16 V (CE)) and 7.9 min & HMVA2 (131® 85 *m/z*, 16 V (CE)), 8.1 min; HICA (131® 85 *m/z*, 16 V (CE)), 8.6 min. The recovery and accuracy were determined during the feasibility step and limits of quantifications (precisions < 15%) were as follows: HMVA – 0.05 μM; HIVA – 0.15 μM; HICA – 0.04 μM. All samples were analyzed in three different batches and, to eliminate the impact of signal attenuation in mass spectrometry, all samples were randomized before analysis. For each batch, blank samples and metabolite standards solutions were injected together with the samples (the calibration curve was carried out at the beginning, the middle and the end of the analytical sequence). The resulting calibration curves gave good linearity with a R2 > 0.99).

**Cell culture and glucose metabolism determinations.** FAO rat hepatocytes were maintained in RPMI 1640 medium (Invitrogen) supplemented with 10% FBS. Cells were maintained in this medium 48 h before treatment. They were then serum-deprived overnight for 16 h with or without insulin (1 nM) and with the indicated doses of HICA, HIVA, HMVA, their combination in a mix, leucine and leucine combined with HICA. The cells were washed three times with PBS and incubated with phenol red- and glucose-free DMEM medium supplemented with 20 mM sodium L-lactate and 2 mM sodium pyruvate for 5 h with or without insulin and the indicated study treatment. Cell supernatant was collected, and glucose concentration was measured with the Amplex-Red Glucose assay kit accordingly to the manufacturer's instructions (Invitrogen). Cells were lysed with 50 mM NaOH, and protein concentration was determined using a BCA protein assay kit to normalize glucose production.

L6 rat myoblasts (kind gift of Dr Amira Klip, Hospital for Sick Children, Toronto, ON, Canada) were grown in alpha-MEM with 10% FBS and differentiated into myotubes in alpha-MEM with 2% FBS. Fully differentiated L6 myotubes were serum-deprived for 5 h in alpha-MEM and treated with the indicated doses of HICA, HIVA, HMVA and their combination in a mix for the last two hours of deprivation. The cells were also stimulated with or without insulin (100 nM) during the last 30 min of deprivation. Glucose uptake was measured in cells incubated for 8 min in HEPES-buffered saline containing 10 μmol/l unlabeled 2-deoxyglucose and 0.33 μCi/ml 2-[1,2-3H(N)]-deoxy-D-glucose (Perkin Elmer). The reaction was terminated by washing three times with ice-cold 0.9% NaCl. Cell-associated radioactivity was determined by lysing the cells with 0.05 N NaOH, followed by liquid scintillation counting and normalization to protein concentration.

**Fecal DNA sequencing and analysis.** Fecal samples were collected in the morning and frozen immediately at −80 °C before and at the end of the dietary treatment. Fecal genomic DNA was extracted using the ZymoBiomics[TM] Miniprep kit and purity was assessed by a nanodrop (Thermo Fisher, USA). Amplification was performed using the V3-V4 primers for 16S rRNA (forward:CCTACGGGNGG CWGCAG, reverse: GACTACHVGGGTATCTAATCC)[80] by Lifesequencing (Valencia, Spain). The samples were loaded into flow cells in an Illumina MiSeq. 300PE Sequencing Platform in accordance with the manufacturer's instructions. Analyses were performed using QIIME (v. 19). Reads were clustered into operational taxonomic units (OTUs; 97% identity threshold) using VSEARCH, and representative sequences for each OTU were aligned and taxonomically assigned using the SILVA database (v. 119). Alpha-diversity (number of observed OTUs, Shannon index, Simpson's reciprocal) and beta-diversity (Bray-Curtis dissimilarity, weighted and unweighted UniFrac distances) indices were computed. The diet effect (H vs C) and the product effect (Y vs H) were evaluated separately. For each type of parameter and each comparison, *p*-values were adjusted for multiple testing considering that several

metrics or genera were tested, using the Benjamini–Hochberg procedure. The alpha risk was set at 0.05 and analyses were performed using R (v. 3.4.0).

Alpha diversity was analyzed using generalized least square models (R packages nlme, emmeans, moments) with baseline, group, study, and group:study interaction taking into account variance differences between groups. The Simpson's Reciprocal index was log10-transformed for normality purposes. Beta-diversity was analyzed using PERMANOVA (R package vegan) for each comparison. The model group included group, time, and group: time effects and the reported p-values are for the group: time effect. Multivariate homogeneity of group dispersion was checked. The DESeq2 R package (with "poscount" option) was used to identify genera differentially abundant between groups experiment-wise, with the model study:group:subject.n + study + study:group + study:group:time to take into account the within-group subject effect. Only genera present with at least three counts in three samples were kept for analysis. For each comparison, fold changes were evaluated with Wald tests at the final timepoint.

**Quantification of *S. thermophilus* CNCM I-1630 strain by qPCR.** The strain was quantified in cecal (12 mice per group for Study 1) and fecal samples (24 mice per group for Study 1 and 21 mice per group for Study 2). Genomic DNA was extracted from 9 mg of fecal material using Fast DNA spin kit for feces (MP biomedicals) and the strain count was assessed using a real-time quantitative PCR (qPCR, Genalyse Partner) using strain-specific primers and probe (Forward primer: 5′-GAAGAAATCGCTAAAGCAGGTATTAAAG-3′, Reverse primer: 5′-AATTCTTAATGACGGAAATGTCTCACG-3′ and probe: 5′ FAM-AGTACGT GACAAAGAGTCAATCGAAGCGG-3′ Iow BlackR® FQ). The qPCR run was performed in duplicate for Study 1 and in triplicate for Study 2 by capillary LightCycler 480 system (Roche). qPCR reactions included LC480 probe mastermix (Roche), 200 nM each forward and reverse primer, 100 nM target specific Taqman® probe (100 nM final), and 2 μL DNA. The qPCR amplification was programmed with an initial denaturation at 95 °C for 10 min, followed by 45 cycles of 95 °C/15 s and 62 °C/15 s. A genomic DNA standard for qPCR assays was obtained from pure targeted strains and composed of 6 successive dilutions from $10^6$ to 10 copies of genomic DNA.

**Hepatic and fecal bile acids measurement.** Lyophilized feces samples (5 mg) were homogenized using a bullet blender (Next Advance, NY, USA) and 100 mg of stainless-steel beads (0.9–2 mm) in 500 μL of a water:methanol (50:50) containing 0.1% formic acid solution. Fifty microliters of internal standard (mix of CDCA-d4, DCA-d4, CA-d4, LCA-d4, TCA-d5 and GCA-d4; C/D/N Isotopes Montréal, Canada) were added. Homogenates were centrifuged at $5000 \times g$ for 5 min. Supernatants were then evaporated under nitrogen and resuspended in 1 mL of a water-0.1% formic acid solution, before performing solid-phase extraction (SPE) using a pre-conditioned (methanol and water-0.1% formic acid) Strata-X 60 mg 96-wells plate (Phenomenex, Torrance, CA, USA). SPE columns were washed with water (2 mL) and a water: methanol solution (80:20) containing 0.1% formic acid (2 mL). Analytes were then eluted using 2 mL methanol. Elutes were evaporated under nitrogen and reconstituted in 100 μL water: methanol (50:50) prior injection to the LC-MS system. The same procedure was also applied to analytical standards initially diluted in adsorbed plasma. One μL of samples or analytical standards was then injected into the chromatographic system consisting of a Nexera ultra-high-pressure liquid chromatography instrument (Shimadzu Scientific Instruments, Columbia, MD, USA). The chromatographic separation was achieved with a C18 column from Agilent (150 × 2.1 mm Poroshell 120 EC-C18; 2.7 μm particles; Santa Clara, CA) at 45 °C, and the following mobile phases: solvent A = ammonium acetate in water (10 mM) at pH 7.0 and solvent B = acetonitrile. Separation was performed at a flow rate of 0.35 mL/min using the following sequence: 85% A:15% B as initial conditions immediately increased to 18% B in 0.5 min and held for 2.5 min, then a linear gradient to 33% B over the next 30 min, followed by an increase of B to 56% in 8 minutes. The column was then flushed at 95%B over the next 12 min and back to initial conditions for 8 min. All analytes were quantified by tandem mass spectrometry (MS/MS) using an API6500 instrument (Applied Biosystems, Concord, ON, Canada). The temperature was set at 500 °C.

**Statistical analysis.** As the mice were single caged housed, the experimental unit for all parameters was the mouse. For each quantitative parameter, a descriptive analysis was conducted by group (diet or treatment) and by group and timepoint. Except if stated otherwise, the statistical strategy was (1) to assess the effect of the H diet as compared to C diet and then, in case of a significant effect (2) to test the Y treatment impact on attenuating the H diet effect. All tests were performed by a parameter; the alpha risk was set at 0.05 (two-sided), except if stated otherwise; a gatekeeping strategy was performed within the (1) and (2) but not between. For linear models, homogeneity of variance was assessed using Levene tests (alpha = 0.05) and normality of the residuals was assessed using Shapiro-Wilk tests (alpha = 0.01). If skewness and kurtosis were in the range]−1.5,1.5[, normality was still considered as acceptable. Liver qPCR data were analyzed as delta-delta Ct for normality purposes. Depending on the parameter, R and SAS 9.4 were used to generate the statistical analyses. The following models were used depending on the type of measurements: (i) for quantitative parameters measured at one-time point a one-way analysis of variances (ANOVA) with a fixed group effect (diet or

treatment) and a random batch effect were used. If the hypothesis of normality was not met, original data were log10 transformed; (ii) for quantitative parameters measured at two-time points, the baseline effect was first checked as for parameters measured at only one-time point. Then the second time point was analyzed using a one-way ANOVA with a fixed group effect adjusted by baseline value for (1) and (2). If the hypothesis of normality was not met, non-parametric statistics were used on variations versus baseline: Mann–Whitney tests for (1) and (2); (iii) for quantitative parameters measured at more than 2 timepoints except energy intake, the baseline effects were first checked as for parameters measured at only one-time point. Then the remaining timepoints were analyzed using a two-way ANOVA with repeated measures with fixed effects group, time, and time*group interaction and adjusted for baseline value. If the hypothesis of normality was not met, non-parametric Mann–Whitney tests were used on variations vs. baseline.

For bile acids, cytokines/chemokines, and liver qPCR, p-values were corrected for multiple testing using a Benjamini–Hochberg adjustment. For HIVA, HICA, and HMVA, T-tests were performed for each tissue independently, to address (1) and (2), but (2) was assessed, even if (1) was not significant, in an exploratory way. P-values were adjusted for multiple testing considering all comparisons performed for one tissue using the Benjamini–Hochberg procedure. Pearson correlations were assessed between BCHA (HIVA, HICA, HMVA) using log-transformed values of metabolomics intensity and hepatic triglycerides, fasting insulin, and fasting glucose for Study 1 and 2. P-values were adjusted for multiple testing for each parameter using the Benjamini–Hochberg procedure.

For the GF Study, a t-test or the corresponding nonparametric Mann–Whitney test was performed, while significant differences for repeated measures were detected using a mixed model followed by a Tukey post hoc test (GraphPrism v8.0.1).

For cells experiments, a one-way analysis variance (ANOVA) was performed, followed by Dunnett post hoc test. If data did not respect normality and homoscedasticity postulates, the non-parametric equivalent Kruskal–Wallis followed by Dunn's post hoc test was applied. For quantitative parameters measured at more than one-time point (HGP of FAO cells treated with two concentrations of HICA, leucine and their combination), data were analyzed using a two-way ANOVA with fixed factors group, dose, and group*dose, followed by Dunnett post hoc test (GraphPrism v8.0.1).

**Reporting summary**. Further information on research design is available in the Nature Research Reporting Summary linked to this article.

## Data availability

The data that support the findings of this study are available from the corresponding author upon reasonable request; the raw data of 16S rRNA sequence for the project through the European Nucleotide Archive (https://www.ebi.ac.uk/ena/) under accession number PRJEB47834 and the raw metabolomics data from MetaboLights (https://www.ebi.ac.uk/metabolights/) under study number MTBLS442

## Code availability

No custom code was used in this work.

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

## Acknowledgements

We gratefully thank Christine Dion, Valérie Dumais, Christine Dallaire, Joanie Dupont-Morissette, Jacinthe Julien, Jenny Rancourt, Cinthia Faubert and Alexandre Pleau for their help with animal care, and Perrine Feutry for SCFA measurement. We also thank Agnes Cartier Meheust for her input on the nutritional profile of diet, and François Debru, Luc Terragno, Fanny Larerre for their input on the design, development, and production of study products. We thank Celine Rio, Lauriane Raidot, Jean Miguet for their data management, Muriel Derrien for her bioinformatics treatment of fecal DNA sequences, Anne Druesne, Christele Tison for their management of study product, nutritional analysis, microbiological analysis, and preclinical management, Hue Del-homme, Guillaume Gobert for qPCR support, Angélique Volpe, Nicolas Bouchet for overall analytical support. Jean-Michel Faurie and Christian Chervaux for their input on microbial metabolic pathways. Research relating to this study was funded through a research collaboration agreement by Danone Nutricia Research Palaiseau, France and the IUCPQ institute-Laval University, Quebec, Canada. Part of the research was also funded through a Canadian Institute of Health Research (CIHR) grant to A.M. (FDN-143247). A.M. holds a CIHR/Pfizer research Chair in the pathogenesis of insulin resistance and cardiovascular diseases. The lipidomics CLP & metabolomics HD4 was funded by Danone Nutricia Research, Palaiseau under service agreement with Metabolon, USA. The targeted NMR and LC-MS/MS analysis were funded by Danone Nutricia Research, Palaiseau under service agreement with Bioaster, France.

## Author contributions

N.D., H.K., and A.M. designed the diets and animal studies. A.M. and H.K. designed the in vitro studies. H.K. and A.M. supervised the research. T.T.T. contributed to the conduct and interpretation of the first two animal studies. N.D. and R.N. carried out animal studies and interpreted the results. A.O. and B.M. carried out the in vitro studies. M.J.D. performed the QPCR analysis and interpreted the resulting results. P.s.P. and R.N. contributed to the design and interpretation of the clamp study. A.G. and P.J. performed liver histological analyses. J.T. and O.B. performed B.A. analyses and interpreted the data. H.K., A.C., and L.Q. designed the targeted lipidomics and untargeted metabolomics profiling study and H.K., M.P. interpreted data. H.K. designed and interpreted the targeted NMR and LC-MS/MS study. Statistical analyses were handled by A.C. and L.Q. for animal studies; M.P., M.S., and T.V. for the fecal microbiota analyses; N.D. and R.N. for in vitro studies supervised by L.Q.; M.P. for LC- MS/MS and M.P. validated the targeted

lipidomics and metabolomics profiling. N.D., R.N., H.K., and A.M. wrote the manuscript. All authors edited the manuscript.

## Competing interests

A.C., L.Q., H.K. are, and T.T.T. was employees of Danone Nutricia Research (DNR), Palaiseau, France. At the time of the study, M.P. was an employee of IT&M Innovation on behalf of DNR, Palaiseau, France, and M.S. was an employee of Soladis on behalf of DNR, Palaiseau, France. DNR provided the test yogurt product and has filed patent applications based on the enclosed findings. The remaining authors declare no competing interest.
