## [Peer Review File · Nature Communications]

REVIEWER COMMENTS

Reviewer #1 (Remarks to the Author):

The present study describes the impact of yogurt on rodent models of diet-induced obesity. Here, the authors link branched chain hydroxy acids to improved insulin action, and further characterized the contribution of the gut microbiota to metabolic improvement. This work established a new nutritional, and potential therapeutic application through branched chain hydroxy acids. The authors should be commended for the approach to investigating the tissues and mechanisms of insulin modulation.

As the authors note, the impact of yogurt consumption on insulin in a different mouse model of insulin resistance (REF #19). It is significant that yogurt had the above effects in a DIO model, but raises the question how model-specific this activity is. Certainly, data from experiments conducted in cultured liver and muscle cells (Fig. 5) support the hypothesis that the mechanism is reproducible under other conditions.

It is likewise notable the model induced metabolic endotoxemia and liver inflammation, but these were not robustly inhibited by yogurt supplementation.

In my opinion, this manuscript is very well-written, but could be improved by addressing the following:

- 1) The composition of yogurt can vary in its bioactive component(s). The authors have presented a thorough compositional analysis, but would help to more clearly define the process used to produce the yogurt. Were these fermented in-house? For how long? Were they stored after fermentation or frozen directly? What was the initial heat treatment? I'm assuming bovine milk here.... Why use these particular strains? Were cultures viable after lyophilization? What was the exopolysaccharide content/profile (if relevant)?
- 2) The protein profile of the dietary interventions is interesting, but not widely used in the field. What implications would the composition have on the abundance of BCHA in this model? Some more information on BCHA precursors (leucine, isoleucine, valine) provided by the diets (and in comparison, to standard obesogenic diets employed in the field) would be helpful. Also, in the formulation, how were proteins purified or prepared for the mixture? Do these represent denatured/cooked protein? Ref. 70 establishes the rationale for the formulation, but not the specific methods of preparation.
- 3) I believe it is notable that another group performing metabolomic analysis after fermented milk consumption didn't detect these specific metabolites: <https://doi.org/10.1093/jn/nxy053> and to my knowledge these haven't been picked up in other exploratory biomarker studies:

<https://pubmed.ncbi.nlm.nih.gov/34204298/> in humans. Are there any issues translating these results to human nutritional studies? Is there evidence for BCHA deficiency with obesity? This is partly addressed at LN 378 and following.

Minor comments:

I slightly disagree with the idea that yogurt is “restoring” BCHA – since the experimental approach was based on prevention, rather than treatment of obesity. E.g. yogurt was supplemented throughout the obesogenic phase.

Please define abbreviations in Fig. S1.

Methods: Would help to define the nature of other dairy samples assessed for BCHA – were these commercial samples? Employing similar cultures?

Reviewer #2 (Remarks to the Author):

This paper investigates the hypothesis that yogurt consumption can prevent the development of obesity-linked, type 2 diabetes (T2D) in mice and explores the underlying causes for that outcome. The paper nicely shows how (lyophilized) yogurt incorporated into the mouse chow diet is beneficial for reducing the risk for T2D by preventing insulin resistance and sustaining fasting glucose among other metrics. The mouse studies also provide interesting associations between certain branch chain hydroxy acids (BCHA) in yogurt and elevated levels of these compounds in yogurt-fed mice. The fact that BCHA were also more abundant in the liver of control but not high-fat diet fed mice was notable as were the studies that these compounds are able to modulate liver and muscle cell culture. However, the findings remain associative and therefore the manuscript falls short of providing convincing evidence that BCHAs are the main factors driving how yogurt (and not milk) reduces the risk for type 2 diabetes. Additionally, the authors investigation of the gut microbiota and bile acids suggest other mechanisms for yogurt effects, possibly related to bile acid metabolism. While it remains quite likely that there is more than one pathway via which yogurt benefits health, the pursuit of one (e.g. BCHA or gut microbiome) would have provided for a more compelling paper.

Major concerns:

Despite the descriptions used in the text, the three mouse studies were not focused on obesity treatment, they were instead addressing obesity prevention. This is an important point because if the goal was to examine how yogurt 'improves' (e.g. L111 (and others)) or "restores" (e.g. L128 (and others)) metabolic function, then the experiments should have been performed on obese mice switched to an HFD containing yogurt. Similarly, the paper should be revised to avoid overstatements such as yogurt increased BCHA (L254) when it may simply be that yogurt prevented losses in systemic BCHA levels due to the HFD.

The analysis of the gut microbiota is a weak point in the paper. Old Qiime methods (not supported since 1/2018) were used, relying on the less accurate OTU constructions. Assessments are limited to a single fecal collection time point at the end of study when cecal and ileal contents should have been accessible. Importantly the changes in gut microbiota contents cannot be substantiated because the baseline microbiota was not measured. While increased levels of Streptococcus may make sense in light of the fact that *S. thermophilus* was in the yogurt, this should be validated with qPCR using strain specific primers.

Related to this is that the viability of the bacteria in the lyophilized yogurt and chow should be provided. This is of critical importance to understanding the mechanistic basis for the observations made in this paper.

The gnotobiotic mice study is not sufficiently described in the materials/methods or the results sections. Were fecal slurries given once? Were all mice fed the H diet for 9 weeks? Where are the chow fed controls to illustrate normal OGT and insulin levels in these mice? The missing background and data make it difficult to interpret the extent to which the FMT was responsible for the observed phenotypes.

The crux of the paper is the increase in BCHA in yogurt and in mice consuming yogurt. It is not explained in the text, but Table S4 seems to show that this finding was limited to mouse study 1, and no effect was observed in study 2. The rationale for these differences should be explored. The finding of significant differences between different DIO mice cohorts is not uncommon. Therefore, it is critical to ask how relevant the BCHA results are when they are limited to one out of two (three?) mice studies? To address this, it would be helpful to know the BCHA levels in the intestinal contents and in the gnotobiotic mice.

Related to the last point is that quantification of hepatic gene expression and chemokine/cytokine quantification seems to have been limited to study 1. The lack of effect of yogurt consumption on TG in study 2, strongly indicates that the tissues from study two should also be examined in order to verify that the proposed Y-induced changes in metabolic and inflammatory profiles are not limited to a single study and time point.

L383 Why focus on the indigenous gut microbiome when the more direct effect is the consumption of the bacteria in yogurt with the known genetic capacity to make BCHA? Moreover, it may just be the BCHA in yogurt, without the need for further production in the intestine.

Minor comments:

L54 group should be plural

L55 Unfinished phrase “yogurt has received a particular interest”. The statement should be completed with citation.

L59 Remove the description of the animal study. The other reports described in the paragraph are presumably all human studies. The inclusion of the animal model paper distracts from the overall findings in prospective and RCT work.

L87 Not clear what the “n” stands for here, whether it is the total number of mice or mice/diet group. Revise here and throughout.

Figure S1. The Figure legend is missing. It is needed at least in part to explain the differences between mice in study 3 (subset or different animals entirely for those “analyzed” and examined by HIEC?)

L133 Liver weight in the Y1 group was lower than the lean Controls. The implications for this should be discussed in light of the fact that only the comparison to the H (HFD) fed mice was discussed.

L173-174 Should be stated “between the Y and H fed groups”

L443 A new nomenclature is introduced (H1, H2 groups, etc) that is not supported by explanation in the results text.

L193-203 The wording should be revised to emphasize the distinction between the lipid metabolomics versus the other metabolites identified. It is just stated that “further profiling” was used. This description is too vague and also is not clarified in the Supplemental Tables.

L241-244 How can it be ruled out that the different BCHA have different rates of absorption? The conclusion that there are differences in body BCHA metabolism is indirect and therefore not warranted.

L274 The fact that the mice were consuming equivalent to two servings of yogurt per day should be explained from the start. This was a question throughout the reading of the paper. It is an odd study by the fact that the yogurt was dried and incorporated into the diet and so this too should be explained and made clear to the reader from the introduction.

L290 “that” not “the”

L314-318 Limited to study 1?

L347 This would be at least partially addressed by examining the gut microbiome of the gnotobiotic mice.

Reviewer #3 (Remarks to the Author):

This is a review of a manuscript entitled “The gut microbiota and fermentation-derived branched chain hydroxy acids mediate the health benefits of yogurt consumption in obese mice” authored by Daniel et al. In this manuscript, the authors show that dietary yogurt improves glucose homeostasis and reduces hepatic insulin resistance and liver steatosis in T2D obesity mouse model. The authors also show that yogurt intake alters the liver metabolome particularly increasing branched chain hydroxy acids (BCHA), which correlate with improved metabolic parameters and improve insulin response in glucose metabolism in both liver and muscle tissues.

The manuscript is of great significance, it represents a great contribution to the field and sheds some light on the “dark” area of host-microbiome metabolic interactions. The manuscript is well written, and the data is presented clearly.

Major comments:

1) The authors used several LC-MS methods for the characterization of the signatures of the different samples. Some samples were analyzed by Metabolon while others were analyzed by the authors. In methods section “Lipids and Metabolites Profiling” the authors provide different information and description of their processes starting from sample prep to chromatographic gradient and MS settings for the different analyses. This information is very important to be able to replicate the experiments and the authors need to describe their methods consistently for all their LC-MS analyses and not only a few. Some important parameters are: weight of tissue and volume used for extraction, reconstitution solvent composition, chromatographic gradient, column temperature, QQQ transitions, flow rate, etc. Some of these parameters are described for some methods but not consistently throughout the methods section.

2) Unlike the metabolomic LC-MS analysis methods used by Metabolon, the lipidomics analysis methods were not described at all.

Minor comments:

1) In page 17 line 417, an "S" is missing in the word States.

2) In page 26 line 659, it should say "flushed".

The gut microbiota and fermentation-derived branched chain hydroxy acids mediate the health benefits of yogurt consumption in obese mice
Point-by-point response to the reviewers

Note: The page and line number refer to the manuscript version with apparent corrections.

Reviewer #1:

The present study describes the impact of yogurt on rodent models of diet-induced obesity. Here, the authors link branched chain hydroxy acids to improved insulin action, and further characterized the contribution of the gut microbiota to metabolic improvement. This work established a new nutritional, and potential therapeutic application through branched chain hydroxy acids. The authors should be commended for the approach to investigating the tissues and mechanisms of insulin modulation. As the authors note, the impact of yogurt consumption on insulin in a different mouse model of insulin resistance (REF #19). It is significant that yogurt had the above effects in a DIO model but raises the question how model-specific this activity is. Certainly, data from experiments conducted in cultured liver and muscle cells (Fig. 5) support the hypothesis that the mechanism is reproducible under other conditions.

It is likewise notable the model induced metabolic endotoxemia and liver inflammation, but these were not robustly inhibited by yogurt supplementation.

Response: We thank the reviewer for the positive comments and for providing important feedback on our manuscript. We agree that the impact of yogurt intake on insulin sensitivity may vary according to the animal model used. In ref 30 (Johnson *et al.*) the authors used a genetic model (e.g., the F1 generation of a genomic cross between BTBR and C57BL6 mice) that is suitable to test the role of genetic factors in the development of insulin resistance, but which is less appropriate to determine the impact of dietary interventions on insulin resistance and type 2 diabetes. We have carefully selected our dietary model to not only reproduce the typical higher consumption of fat and sugar that are well known to promote obesity and insulin resistance, but also made sure the animals were consuming dietary proteins that are typically consumed by humans instead of only casein as still mostly used in pre-clinical studies in rodents.

In my opinion, this manuscript is very well-written, but could be improved by addressing the following:

1) The composition of yogurt can vary in its bioactive component(s). The authors have presented a thorough compositional analysis but would help to more clearly define the process used to produce the yogurt.

Were these fermented in-house?

Response: For all studies, the yogurt was produced at the Danone Nutricia Research site.

For how long?

Response: The yogurt was fermented at 39°C for 9h.

Were they stored after fermentation or frozen directly?

Response: The yogurt was stored at 4°C for 2-3 days before lyophilization. The lyophilized yogurt was stored at 4°C.

What was the initial heat treatment?

Response: The milk was pasteurized at 95°C for 5 min.

I'm assuming bovine milk here....

Response: The yogurt was indeed produced using the bovine milk.

Why use these particular strains?

Response: A fermented milk can only be called "yogurt" if it is fermented with *Streptococcus salivarius subsp. thermophilus* (*S. thermophilus*) and *Lactobacillus delbrueckii subsp. bulgaricus* (*L. bulgaricus*) (EFSA, *EFSA journal*, 2010, doi: 10.2903/J.efsa.2010.1763) leading to bacterial protooperation (Arioli, *International Journal of Food Microbiology*, 2017, doi: 10.1016/j.ijfoodmicro.2016.01.006). Also, the yogurt was selected for its representative macronutrient composition (un-supplemented with calcium, vitamin D, starch, no fruits or flavors) and the two particular strains CNCM I-1519 (*L. bulgaricus*) and CNCM I-1630 (*S. thermophilus*) as it represents an analytical advantage (microbiology, qPCR) over yogurts fermented with multiple *S. thermophilus* and / or *L. bulgaricus* strains.

We thank you for these valuable points above and we integrated details on process used to produce the yogurt in the Methods (see p.18, L444-445).

Were cultures viable after lyophilization?

Response: Thank you for this question. Yes, the cultures were viable after lyophilization: the microbiological composition of lyophilized yogurt was determined at production and stability was checked over time using plate count method (ISO 7889:2003 IDF 127 Yogurt-Enumeration of characteristic microorganisms). We have integrated all the details of yogurt productions and microbiological profiles in the Methods (see p.19, L466-468) and Table 2.

What was the exopolysaccharide content/profile (if relevant)?

Response: The exopolysaccharide profile is related to the final biomass, i.e., *S. thermophilus* strain CNCM I-1630 in the tested yogurt. Using the phenol-sulfuric method for the total exopolysaccharide quantity and after extraction and acidic hydrolysis, the composition of the exopolysaccharide building-block was determined using the HPLC as rhamnose/galactose in a ratio 1:2 (data not shown). The ramification, size, and chain modification (acetylation, methylation, phosphorylation) were not specifically determined as exopolysaccharide quantification is fastidious, limited by the extraction yield and complex due to large diversity of produced exopolysaccharides. It is unclear at present if the exopolysaccharide content/profile is relevant to the beneficial effects of yogurt.

2) The protein profile of the dietary interventions is interesting, but not widely used in the field. What implications would the composition have on the abundance of BCHA in this model? Some more information on BCHA precursors (leucine, isoleucine, valine) provided by the diets (and in comparison, to standard obesogenic diets employed in the field) would be helpful. Also, in the formulation, how were proteins purified or prepared for the mixture? Do these represent denatured/cooked protein? Ref. 70 establishes the rationale for the formulation, but not the specific Methods of preparation.

Response: We thank the reviewer for this interesting question. We compared the levels of leucine, isoleucine and valine (the branched-chain amino acids, BCAA) in several diets, used in 1) our recently published study (Choi, Daniel *et al.* -ref 22-) comparing the replacement of casein by a protein mix in high-fat high-sucrose (HFHS) diet, 2) the classic D12492 high-fat diet marketed by Research Diet (casein only as protein source) and 3) our current study in H and Y diets, respectively (Table below):

Source	Choi, Daniel et al. (Ref.22)	Choi, Daniel et al. (Ref.22)	Research Diet	Current Study	Current study
Diet	HFHS	HFHS	D12492	H	Y
Protein source	Casein	Protein mix	Casein	Protein mix	Protein mix
%kcal of protein in diet	15	15	20	15	15
Leucine (g/100g of diet)	1.4	1.5	2.0	1.6	1.5
Isoleucine (g/100g of diet)	0.7	0.8	1.0	0.9	0.9
Valine (g/100g of diet)	0.9	1.1	1.2	1.0	1.0
Total BCAA (g/100g of diet)	2.9	3.3	4.2	3.5	3.4

Table: Comparison of BCAA levels in Choi, Daniel *et al.* publication, D12492 diet from Research Diet, and H and Y diets used in the current study.

A sentence was added in the Methods section in the “Dietary treatments” subsection (see p.19, 486-488):

“Branched-chain amino acid content (BCAA) was analyzed and equivalent to 3.5g/100g of diet for H diet (1.6g/100g leucine, 0.9g/100g isoleucine, 1.1g/100g valine) and 3.4g/100g of diet for Y diet (1.5g/100g leucine, 0.9g/100g isoleucine, 1.1g/100g valine).”

We found in our study that H and Y contained the same level of BCAA (isoleucine, leucine, valine), the precursors of the BCHA. Thus, the BCAA content in diets could not have impacted the abundance of BCHA in our animal model, further suggesting that the difference in BCHA in the experimental groups may come from yogurt starter activity, and/or from different microbial and/or host metabolism of BCAAs (this sentence was added in the discussion, see p.15-16, L385-387). Also, we found that there is slightly more BCAA in D12492 (mostly explained by the higher protein proportion) compared to our diet and the same range in BCAA levels in HFHS diets (Choi, Daniel *et al.*).

As for protein sources used in the diet, all of them were bought already cooked and provided in lyophilized form: beef, pork, and chicken (Happy Yak®, Canada), soy and white egg (Teklad Envigo, USA), casein (MP biomedical, Canada). The nutritional profile of each source was analyzed, and the protein mix consisted of 82%g protein, 0.5%g carbohydrate, and 13%g fat. As it was complicated to delipidate the protein sources, we chose to analyze nutritional composition and carefully match diets for 1) the protein 2) the lipid 3) the carbohydrate and 4) the saturated: polyunsaturated fatty acid ratio. These details were added in the Methods section (p19, L474-482). For further details, please refer to Choi, Daniel *et al.* 2021.

3) I believe it is notable that another group performing metabolomic analysis after fermented milk consumption didn’t detect these specific metabolites: <https://doi.org/10.1093/jn/nxy053> and to my knowledge these haven’t been picked up in other exploratory biomarker studies: <https://pubmed.ncbi.nlm.nih.gov/34204298/> in humans. Are there any issues translating these results to human nutritional studies?

Response: Thank you for your interesting comments and for sharing highly relevant publications. There are some experimental differences that could explain why these metabolites were not detected.

Li et al. (<https://pubmed.ncbi.nlm.nih.gov/34204298/>) analyzed targeted metabolite panel of 37 previously identified food intake biomarkers of yogurt consumption and BCHA were, unfortunately, not part of it.

As for Pimentel *et al.* (<https://doi.org/10.1093/jn/nxy053>), they used a different platform from the one of Metabolon and large number of metabolites remained of unresolved chemical structure leaving open the possibility that some of them could be assigned to BCHA. Indeed, they used an untargeted approach to analyze human serum following two weeks of yogurt consumption while we detected differential abundances of BCHA in livers of mice fed with yogurt for 12 weeks. Using a targeted approach, we detected differential BCHA concentrations in plasma, yet the magnitude of the change was less pronounced in plasma than what we observed in liver biopsies.

Moreover, they tested yogurt fermented with classical yogurt starter cultures *L. bulgaricus* and *S. thermophilus* together with *Lactobacillus rhamnosus Gorbach-Goldin* (LGG) and the role of probiotics LGG strain on BCHA production remains to be determined. In our study the skyr product (Fig. 4D) contains 11.3 mg/L of BCHA and is fermented with *L. bulgaricus*, *S. thermophilus* but also *Lactobacillus rhamnosus* and *Lactococcus lactis lactis*. For comparison, Yogurt 2 contains 13.6 mg/L of BCHA and is fermented with *L. bulgaricus* and *S. thermophilus* only. The skyr BCHA are lower yet the skyr protein is more than doubled of Yogurt 2 (10g versus 3.8g /100g of product) suggesting that protein content (BCAA source) is not the only factor determining BCHA production but other factors such as lactic acid bacteria activity and/or technological process likely play a role.

In our study, the Methods to measure BCHA was based on the protocol developed by Ehling *et al.* (now part of the Methods section) (<https://pubmed.ncbi.nlm.nih.gov/24495238/>) to quantify α -hydroxyisocaproic acid (HICA) in milk and plain low-fat yogurt (purchased in the Columbus, OH, metropolitan area). In milk, HICA was below level of quantification (5 μ g/L). However, in four commercial yogurts tested, HICA varied from 3.0 to 15.2 mg/L range, corroborating our data, Fig. 4D.

We agree with the Reviewer that translating the results of these animal investigations to human nutritional studies will be important, and we hope our findings will spark great interests in investigating the role of BCHA in the regulation of human metabolic diseases.

Is there evidence of BCHA deficiency with obesity? This is partly addressed at LN 378 and following.
Response: Thank you for the relevant question. Our data are new and, to the best of our knowledge, the first to show BCHA deficiency in a rodent model of obesity and will need to be confirmed in human obesity in future studies.

Minor comments:

I slightly disagree with the idea that yogurt is “restoring” BCHA – since the experimental approach was based on prevention, rather than treatment of obesity. E.g., yogurt was supplemented throughout the obesogenic phase.

Response: We agree with the reviewer and have replaced “restoring” by “preventing” or equivalent terms throughout the paper.

Please define abbreviations in Fig. S1.

Response: The abbreviations were defined in the new developed legend of Fig. S1.

Methods: Would help to define the nature of other dairy samples assessed for BCHAs – were these commercial samples? Employing similar cultures?

Response: All dairy products tested were purchased commercially at the supermarket in the Paris region. Yogurts 1 and 2 are fermented with *L. bulgaricus* and *S. thermophilus* yet they differ at strain level and macronutrient composition. Yogurt 3 symbiosis remains undisclosed. Fermented Milk 1 is fermented with *L. bulgaricus*, *S. thermophilus*, *Bifidobacterium animalis lactis* and *Lactococcus lactis lactis*. Fermented Milk 2 is fermented with *L. bulgaricus*, *S. thermophilus* and *L. casei*. The skyr is fermented with *L. rhamnosus*, *Lactococcus lactis lactis*, *L. bulgaricus* and *S. thermophilus*. We integrated nutritional and symbiosis details of cultures in a new supplementary figure (Fig. S6) and added a referring sentence in the Methods (see p.19, L468-470).

Reviewer #2:

This paper investigates the hypothesis that yogurt consumption can prevent the development of obesity-linked, type 2 diabetes (T2D) in mice and explores the underlying causes for that outcome. The paper nicely shows how (lyophilized) yogurt incorporated into the mouse chow diet is beneficial for reducing the risk for T2D by preventing insulin resistance and sustaining fasting glucose among other metrics. The mouse studies also provide interesting associations between certain branch chain hydroxy acids (BCHA) in yogurt and elevated levels of these compounds in yogurt-fed mice. The fact that BCHA were also more abundant in the liver of control, but not high-fat diet fed mice was notable as were the studies that these compounds are able to modulate liver and muscle cell culture. However, the findings remain associative and therefore the manuscript falls short of providing convincing evidence that BCHA are the main factors driving how yogurt (and not milk) reduces the risk for type 2 diabetes. Additionally, the authors investigation of the gut microbiota and bile acids suggest other mechanisms for yogurt effects, possibly related to bile acid metabolism. While it remains quite likely that there is more than one pathway via which yogurt benefits health, the pursuit of one (e.g., BCHA or gut microbiome) would have provided for a more compelling paper.

Response: We thank the reviewer for the constructive feedback on the value of our work. While we agree that our studies remain associative for the role of BCHA in the yogurts effect in obese mice, we have provided *in vivo* evidence that the gut microbiota is causally implicated by performing FMT studies in germ-free mice. Furthermore, we have showed that BCHA can directly modulate glucose metabolism in liver and skeletal muscle cells providing evidence for the cell-autonomous impact of these molecules on key metabolic targets. The next step will be to directly test the *in vivo* effect of BCHA in animals, but this is currently challenging because we know so little about their metabolic fate and especially their levels in the portal circulation where they are released after yogurt exposure and/or upon microbial metabolism of dietary BCAA. We also suspect that administrating these molecules directly in the systemic circulation may impact the CNS or other systems. We are thus considering testing the metabolic impact of BCHA using a liver perfusion model to prevent their potential confounding effects in the CNS. This will require us to establish new Methods and we thus hope the reviewer will agree that these studies, although of obvious interest, are beyond the scope of this study.

Major concerns:

Despite the descriptions used in the text, the three mouse studies were not focused on obesity treatment, they were instead addressing obesity prevention. This is an important point because if the goal was to examine how yogurt 'improves' (e.g., L111 (and others)) or "restores" (e.g., L128 (and others)) metabolic function, then the experiments should have been performed on obese mice switched to an HFD containing yogurt.

Response: We agree with the reviewer and have changed the terminology and used terms referring to "prevention" rather than "treatment" throughout the paper.

Similarly, the paper should be revised to avoid overstatements such as yogurt increased BCHA (L254) when it may simply be that yogurt prevented losses in systemic BCHA levels due to the HFD.

Response: Thank you for the relevant point. We agree and have modified the text.

The analysis of the gut microbiota is a weak point in the paper. Old Qiime Methods (not supported since 1/2018) were used, relying on the less accurate OTU constructions.

Response: We agree with the Reviewer that the Qiime Methods has been recently replaced by more robust approaches (such as amplicon sequence variant (ASV) or sub-operational-taxonomic-unit (sOTU)). A large part of our analysis is, however, based on genus aggregation data allowing us to interpret the results with an acceptable level of confidence, despite the use of OTU method. Most importantly, the *Streptococcus* increase was further confirmed by qPCR in cecal and fecal contents, and results presented in a new Supplementary Figure 4.

Assessments are limited to a single fecal collection time point at the end of study when cecal and ileal contents should have been accessible. Importantly the changes in gut microbiota contents cannot be substantiated because the baseline microbiota was not measured.

Response: Thank you for the comment and we agree that baseline microbiota is important to substantiate the changes in gut microbiota. We respectfully bring to the Reviewer's attention that Study 1, 2 and 3 had baseline microbiota measured as reported with the sentence: "*Fecal samples were collected in the morning and frozen immediately at -80°C before and at the end of the dietary treatment.*" We are sorry if this was not made clear enough throughout the manuscript and we have clarified the FFM abbreviation (Fresh Feces collection for Microbiota analysis) in Figure S1 indicated the timepoints of feces collection. Thus, all our analyses took baseline microbiota into account.

As mentioned in the Methods part (see p.36, L919-921) for Fecal DNA Sequencing and analysis, alpha-diversity models were adjusted to baseline, and what is plotted in Figure 3A are the means of the linear model, i.e., values already adjusted by baseline.

For beta-diversity, we have now clarified more specifically in the Methods part (see p.36, L922-924) that the permanova models were performed for each experiment with group, time, and group: time effects. The reported p-values are for the group: time effect, which highlights differences in groups that are time-dependent, and thus not driven by potential differences in baseline microbiota between groups. These precisions were added: "*The model group included group, time, and group: time effects and the reported p-values are for the group: time effect.*"

For genera analysis, as mentioned in the Methods part (see p.36-37, L924-927), DESeq2 models considered a subject effect, as recommended by the R package developer

(<https://support.bioconductor.org/p/67457/>), and thus results should also not be driven by baseline differences.

Regarding cecal and ileal contents, we have only collected them in Study 1. The caecal content was dedicated to the short chain fatty acids measurement and qPCR analysis of *S. thermophilus* CNCM I-1630 (Supplementary Table 1 and Figure S4). However, in an attempt to respond to the reviewer's request, we have explored the possibility of determining ileal microbiota, and obtained the following results:

Figure: 16S rRNA sequencing of mice ileal content issued from Study 1. (A) Shannon index and (B) Simpson's reciprocal index. DESeq log2 fold change analysis for (C) C1- vs H1-fed mice and (D) H1- vs Y1-fed mice. (E) Heatmap representing the relative abundance of *Streptococcus* genus in the three groups of treatment. n=7, 9 and 6 for the C, H, and Y groups, respectively.

We observed no significant differences between groups in alpha-diversity (A and B) indices. However, the DESeq log2 fold changes analysis at the genus level revealed that ileal microbiota was enriched in *Turicibacter* and *Bacteroides* genera in C-fed mice compared to H-fed mice (C). Interestingly, this loss of *Turicibacter* was prevented in Y1-fed mice, and we also found that *Streptococcus* genera was more abundant in this group compared to H1-fed mice (D). However, despite this homogenous presence of this bacterium of interest in the Y1-fed group (E), the number of animals in which we were capable of analyzing the ileal microbiota was limited. We performed the analysis on 7, 10 and 10 mice in the C1, H1 and Y1 groups respectively but DNA quality was not satisfying (260/280 ratio between 1.60 and 1.80, 260/230 between 0.4 and 2.11 with an average of 1.3). Among those, we sequenced respectively 7, 9 and 6 ileal content samples from C1, H1, Y1 and obtained the above results. We think that the low DNA quality might come from the difficulty we had to collect ileal content during euthanasia, with a lack of material probably due to the fasting state of the animals. Moreover, ileum sample often contained low bacterial biomass and a large amount of contaminated host DNA in comparison with fecal sample, which explained why 16S rRNA-based amplification may have had very low yields and thus resulted in amplicon not suitable for sequencing. Another limitation of these data is that we could obviously not have access to baseline ileal samples, unlike for the fecal microbiome analyses. Given these limitations, we have elected not to include these data in the current version of the revised manuscript, but we are happy to share them with the reviewers and editors for full transparency.

While increased levels of *Streptococcus* may make sense in light of the fact that *S. thermophilus* was in the yogurt, this should be validated with qPCR using strain specific primers.

Response: Thank you for the comment and we agree that the qPCR validation using strain specific primers is relevant. We quantified *S. thermophilus* CNCM I-1630 by qPCR method in fecal samples collected at the start and end of feeding (e.g., 12 or 15 weeks). The *S. thermophilus* CNCM I-1630 was detected at the level of about 9 log₁₀/g of feces (logarithm of cell number; 8.76 log₁₀/g in Study 1 and 8.86 log₁₀/g in Study 2) in Y fed mice while it was not detected in most animals in the H fed mice or baseline Y fed mice (level of quantification 4.2 log₁₀/g in Study 1 and 4.3 log₁₀/g in Study 2).

“The S. thermophilus CNCM I-1630 was quantified in fecal material and total cell count reached 9 log₁₀/g feces or caecum after yogurt feeding (Fig. S4).”

We integrated a new paragraph for this measure in Methods (see p.37, L931-943) and the Acknowledgements (see p.45, L1230-1231) accordingly.

Related to this is that the viability of the bacteria in the lyophilized yogurt and chow should be provided. This is of critical importance to understanding the mechanistic basis for the observations made in this paper.

Response: Thank you for the comment that was also raised by Reviewer 1. The microbiological composition of lyophilized yogurt was determined at production and stability was checked over time using plate count method. We integrated details on microbiological profile in Methods (see p.19, L466-468) and Table 2.

The lyophilized yogurt was manually incorporated in the H diet every day and all the diets were changed daily prior to the onset of the dark cycle. Therefore, we ensured the viability and count stability of the bacteria in Y diet at least over 20h at 22°C before the Study 1 start, see below.

Strain (Log10 CFU/g)	Time 0h - formulation	Time 16h from formulation	Time 20h from formulation
S. thermophilus	7.9	8.0	8.0
L. bulgaricus	6.1	5.9	5.8

The gnotobiotic mice study is not sufficiently described in the materials/Methods or the results sections.

Were fecal slurries given once?

Response: Yes, mice were gavaged only once with fecal slurries. Chassaing *et al.* (Chassaing, *Gut*, 2017, doi: 10.1136/gutjnl-2016-313099) showed that a single gavage is sufficient to establish and maintain the transferred microbiota in isocage system conditions. A sentence was added in the Methods section (see p.21, L516-517) to precise this point.

Were all mice fed the H diet for 9 weeks?

Response: Yes, and this has been further clarified in the text:

“To explore the role of the gut microbiota in the beneficial effects of yogurt treatment in H-fed 192 mice, we next performed fecal microbiota transplantation (FMT) studies where germ-free (GF) mice were 193 inoculated with fecal slurries recovered from H-fed or Y-treated animals from Study 1 and fed H diet, denominated as H1-T and Y1-T mice.” (See p.8, L192-195).

And in the Methods.

“One week before fecal material transplantation and for all the 9-week duration of the protocol, GF mice were fed with H diet previously supplemented with 50% more vitamins and irradiated twice (29.1-41.6 kGy, Nordion, Laval, QC).” (See p.20, L494-496)

“Mice were maintained in the GF facility 517 and fed the H diet for nine weeks.” (See p.21, L517-518)

This information was also inserted in the experimental design (Figure S1).

Where are the chow fed controls to illustrate normal OGT and insulin levels in these mice? The missing background and data make it difficult to interpret the extent to which the FMT was responsible for the observed phenotypes.

Response: Thank you for this question. The C diet used in our study is a low-fat low-sucrose diet that differs from standard chow diet. Thus, we felt it was somewhat confusing to show the impact of chow feeding post FMT since our donor mice were fed the low-fat low-sucrose diet. Moreover, the chow diet has been shown to greatly impact the gut microbiota due to its high-fiber content (see Daniel *et al.*, *AJP-Gastrointestinal and Liver Physiology*, 2021; doi: 10.1152/ajpgi.00028.2021). However, to address the reviewer's question we are providing below additional data of GF mice that underwent FMT from donor H1- and Y1- fed mice that were then kept on chow diet for 9-weeks. These data are now part of the manuscript as Supplementary Figure 5:

Supplementary Figure 5. Glucose tolerance test (GTT) of gnotobiotic mice fed with chow diet. (A) glycemic and (B) insulinemic responses of germ-free mice transplanted with feces from H1-fed mice (H1-T-Chow, n=11) and Y1-fed mice (Y1-T-Chow, n=8), during the GTT. Mice were issued from the same cohort than mice used in the current study (main paper) but were fed chow diet and not high-fat. They were fasted 4h before the GTT.

As expected, we observed that glycemic and insulinemic responses are less pronounced in mice fed with chow diet than counterparts fed with H diet (Fig. 4F-G). Moreover, the beneficial impact of the microbiota from yogurt-fed animals was not seen in chow-fed animals, corroborating that yogurt prevents metabolic impairments induced by H diet consumption, but has no impact when mice are fed a healthy chow-diet. These additional data are now provided as Supplementary Figure S5, and we have inserted a sentence about this point in the corresponding part of the results section (See p.8, L200-203).

The crux of the paper is the increase in BCHA in yogurt and in mice consuming yogurt. It is not explained in the text, but Table S4 seems to show that this finding was limited to mouse study 1, and no effect was observed in study 2. The rationale for these differences should be explored. The finding of significant differences between different DIO mice cohorts is not uncommon. Therefore, it is critical ask how relevant the BCHA results are when they are limited to one out of two (three?) mice studies?

Response: We appreciate the Reviewer's comment. We report findings from 3 independent studies. Yogurt treatment was maintained for 12 weeks in Study 1 and 3 but 15 weeks in Study 2. Fortunately, we took fasted blood samples after 12 weeks of yogurt treatment in Study 2 as well so that we could compare at least glucose, insulin, and HOMA-IR values between all three studies and allowing us to pool such data from three independent animal cohorts, as reported in Figure 1. However, we couldn't pool the liver analyses from the three studies. Indeed, livers from Study 3 could not be used for lipids or BCHA levels since they were exposed to radiolabeled glucose and Metabolon platform do not accept such samples. In addition, these livers were collected in hyper-insulinemic conditions because animals were infused with insulin during the clamp to assess insulin sensitivity.

For Study 1 and 2, we have obtained biological replication, but the Y effects did not reach the level of statistical significance in the latter study. We believe that there are some experimental differences that could underlay these differences between studies, especially for the liver phenotypes which could only be measured at the end of these protocols. First, as mentioned above, the duration of the yogurt feeding was 12 weeks in Study 1 and 3 but 15 weeks in Study 2, respectively. The severity of

the metabolic disturbances in H-fed animals may have progressed in Study 2 so that the effect of the yogurt could have been impacted. Another key factor to consider is fasting time. Indeed, the duration of fasting was 4 *versus* 6 hours in Study 1 and 2, respectively. The BCHA absorption may have been reduced upon the extended 6 hours fasting in Study 2 and/or BCHA hepatic metabolism may have been accelerated/modified to preserve anabolic response.

The hepatic alpha-hydroxyisovalerate (HIVA) ratio (comparison Y2 versus H2 group) in Study 2 was 1.14 versus 1.34 in Study 1 and the p-value was 0.103, slightly higher than the statistical threshold in Study 1. When we analyzed the Studies 1 and 2 untargeted metabolomics in a pooled analysis (q-value ≤ 0.05 for H to C comparison; q-value ≤ 0.1 for Y to H comparison), only three metabolites were significantly different and HIVA was one of them (1.24-fold change, q-value 0.081). Furthermore, the Study 2 hepatic BCHA correlates with hepatic triglycerides as shown in a new Figure S6 making us believe that the increase in BCHA in yogurt and in mice consuming yogurt is relevant and reproducible.

We added this new information in the main text (see p.10, L.254), and we integrated the Study 2 results as Supplementary Figure S7.

To address this, it would be helpful to know the BCHA levels:

- in the intestinal contents

We have now measured the BCHA contents in remaining jejunum and colonic samples in the Study 1. The results below are from a low number of samples that we could get from the study and are provided to the reviewers and editor only for the sake of discussion, but we do not feel they can be added to the revised manuscript. In summary:

In the jejunum (n=4 in C1 group, n=7 in H1 and Y1 group), only HIVA was detected, and we observed no difference between groups, due to observed higher variability in those lower number of samples:

The HICA and HMVA levels were below the level of quantification. While it might suggest a higher HICA and HMVA absorption in this section of the small intestine, we can't exclude a technical limitation with the detection of these molecules due to the interference with the intestinal tissue matrix (coming from sample preparation and/or detection method).

We detected BCHAs in colon luminal contents with a high within group variability (n=4 in C1 group, n=5 in H1 group, n=6 in Y1 group). Several samples were below the level of quantification, and we can't exclude either the interference with the colon tissue matrix, the use of low remaining quantity of the material (<30mg) in some samples, or both. The BCHAs appear visually reduced in H1-fed mice versus the C1-fed mice but preserved in the Y1-fed mice, yet we remain cautious about interpreting these data given the low number of valid samples and the observed variability.

We acknowledge that these results are unlikely to explain our main findings of changes in BCHA levels in plasma and tissues between C, H and Y groups. We can only speculate until we have 1) sorted the potential technical limits of measuring these BCHA in gastrointestinal matrices, and 2) establish Methods to study BCHA absorption and production using labeled BCHAs and their BCAA precursors. Clearly, this will take a lot of additional time and effort and are therefore beyond the scope of this paper.

- in the gnotobiotic mice.

Unfortunately, the plasma and tissues that could be collected at euthanasia were very limited in the gnotobiotic mice. Indeed, euthanasia was performed right after the animals left the germ-free sector but also after the glucose tolerance test (with previous 4h fasting). Thus, it was already a long and fastidious experiment to perform within and at proximity to the axenic facility in order to limit the risk of contamination, and we were not able to collect intestinal content for the purpose of measuring BCHA in this compartment.

Related to the last point is that quantification of hepatic gene expression and chemokine/cytokine quantification seems to have been limited to study 1. The lack of effect of yogurt consumption on TG in study 2, strongly indicates that the tissues from study two should also be examined in order to verify that the proposed Y-induced changes in metabolic and inflammatory profiles are not limited to a single study and time point.

Response: We appreciate the Reviewer's comment. We analyzed several hepatic lipid and inflammation-related markers (new panels I and J in Supplementary Figure 3 -panels were also re-ordered in this figure) in mice issued from Study 2 and observed the expected alteration of these markers by the H-diet consumption. However, we did not observe a significant impact of yogurt on these markers, as expected from the lack of significant changes in lipid accretion in the livers of these mice. A sentence was added to the results and the corresponding paragraph revised (see p.7, L161-165). As mentioned above, the difference of Y effect between Study 1 and 2 on liver steatosis is likely explained by the longer duration of yogurt treatment and/or the difference in the fasting periods before euthanasia (4h for Study 1 vs 6h for Study 2).

We also observed higher absolute values for cytokine levels in Study 2 and we believe this may be explained by the fact that we used ELISA in Study 2 (since we only looked at those inflammatory markers that were different in Study 1) while we had used a Multiplex approach in Study 1. In addition, the kit manufacturer was different, which can generate difference in absolute values, as already reported previously (Khan, *et al.*, *Cytometry*, 2004, doi:10.1002/cyto.b.20021).

L383: Why focus on the indigenous gut microbiome when the more direct effect is the consumption of the bacteria in yogurt with the known genetic capacity to make BCHA? Moreover, it may just be the BCHA in yogurt, without the need for further production in the intestine.

Response: We only detected HIVA in jejunum but all BCHAs in the colon (see previous answer). Their molecular ratio in the colon were also different than what we have found in the yogurt, and so we hypothesize that the microbiota has a role to play in their *in vivo* production. However, we did not aim to specifically focus on indigenous gut microbiome, and we did postulate for a direct effect of BCHA provided by yogurt as a key contributor to the effects observed. The yogurt is fermented for 9 hours and measurement of the fermentative kinetics and capacity of yogurt strains versus the other gut bacteria having the genetic capacity to make BCHA through the gastrointestinal tract would be interesting to determine and will be the focus of future research. This has been clarified in the revised paper (see Discussion p.17, L414-417).

“It will be important to determine whether some of the bacterial taxa found to be altered by obesity and yogurt treatment express the enzymes required to metabolize BCAA to BCHA and whether this could contribute to the yogurt effect.”

Minor comments:

L54: group should be plural

Response: We regret this typo error. This has been corrected.

L55: Unfinished phrase “yogurt has received a particular interest”. The statement should be completed with citation.

Response: We regret this unfinished phrase and combined it with the next one to make a clear sentence.

L59: Remove the description of the animal study. The other reports described in the paragraph are presumably all human studies. The inclusion of the animal model paper distracts from the overall findings in prospective and RCT work.

Response: We thank the Reviewer for the suggestion and the animal study description was removed.

L87: Not clear what the “n” stands for here, whether it is the total number of mice or mice/diet group. Revise here and throughout.

Response: This has been clarified (see p.4, L90-95):

“The first two studies (Study 1, n=18-24 per group and Study 2, n=14-24 per group) were carried out to explore [...]. The third study (Study 3, n=36 per group) ...”

Figure S1. The Figure legend is missing. It is needed at least in part to explain the differences between mice in study 3 (subset or different animals entirely for those “analyzed” and examined by HIEC?)

Response: Thank you for the comment. All the mice in each study underwent all the experiments. The difference in n was notably due variations in the number of mice in each group that were able to go through the surgery for canulation and clamp procedure. Some mice, especially in the H-fed obese groups, had their catheter blocked and prevented blood sampling likely due to blood clot formation. As we managed to perform all the non-invasive (or less invasive) tests before the HIEC, the n was only decreased for this more invasive test. We agree with the reviewer that a complete legend was

needed for Figure S1 to clarify the complex experimental design (see revised Supplementary Figure 1).

L133 Liver weight in the Y1 group was lower than the lean Controls. The implications for this should be discussed in light of the fact that only the comparison to the H (HFD) fed mice was discussed. Response: The reviewer is correct, the liver weight of the Y1-fed group appears lower than C1-fed group in panel 2A, but the data are expressed in % relative to body weight. The liver is actually heavier in H1-fed mice compared to that of the controls, and yogurt partly prevented this weight gain (see Figure below when using the raw data and Table S1).

L173-174: Should be stated “between the Y and H fed groups”

Response: We thank the Reviewer for the comment. The modification was made in the text.

L443: A new nomenclature is introduced (H1, H2 groups, etc.) that is not supported by explanation in the results text.

Response: We thank the reviewer for pointing out this typo mistake. The sentence has been corrected as follows in the Methods (see P.20, L511-513):

“For fecal microbiota transplantation (FMT), feces were collected at week 12 in Study 1 from mice fed with H or Y diets”.

L193-203: The wording should be revised to emphasize the distinction between the lipid metabolomics versus the other metabolites identified. It is just stated that “further profiling” was used. This description is too vague and also is not clarified in the Supplemental Tables.

Response: We thank the Reviewer for the comment and apologize for this imprecision. We integrated in the text the clarification about the use of global lipidomic (Complex Lipid Panel, CLP) and metabolomic HD4 platforms in the text. We also modified the Methods (see p.24-35) and Supplementary Tables 2, 3 and 4 headings accordingly.

L241-244: How can it be ruled out that the different BCHAs have different rates of absorption? The conclusion that there are differences in body BCHA metabolism is indirect and therefore not warranted.

Response: We thank the reviewer for the comment. Here, we only suggested a potential explanation raising possible different BCHA rates of absorption. We are sorry that the reviewer perceived the sentence as a conclusion and modified the text to avoid the confusion (see p.11, L269-271) of revised paper:

“...suggesting that the systemic and tissue levels of these metabolites are not only reflecting their nutritional exposure but may also be linked to some BCHA type-specific absorption and/or metabolism.”

L274: The fact that the mice were consuming equivalent to two servings of yogurt per day should be explained from the start. This was a question throughout the reading of the paper. It is an odd study by the fact that the yogurt was dried and incorporated into the diet and so this too should be explained and made clear to the reader from the introduction.

Response: We thank the Reviewer for the comment and indeed this information was described only in the Methods under “Dietary treatments”. We now better explain that the yogurt was dried and incorporated into the diet daily which was necessary to provide enough product to reach a dietary equivalent to two servings of yogurt per day as follows in Introduction (see p.3, L74) and at the start of Results (see p.4, L88-90) of revised paper.

L290: “that” not “the”

Response: This was modified.

L314-318: Limited to study 1?

Response: We thank the Reviewer for this question. The L314-318 generally refers to all three studies and following sentence refers to Study 1 and 3 but not Study 2 as underlined below (see p.14, L350-356).

“Here, we report that yogurt treatment of H-fed mice induced a slight reproducible shift in the gut microbiota in all three studies with an enrichment of *Streptococcus* and a depletion of an unknown *Peptostreptococcaceae*. We further observed a few study-specific changes in bacteria from the Firmicutes phylum. In Studies 1 and 3, where we saw more robust effects of the yogurt treatment, the metabolic improvements were further associated with a lower relative abundance of bacteria from *Peptococcaceae* or *Lachnospiraceae Incertae Sedis* families, as compared to H-fed controls.”

L347: This would be at least partially addressed by examining the gut microbiome of the gnotobiotic mice.

Response: We agree with the reviewer, but we inoculated several pools of donor feces to the germ-free animals, so we have a more limited resolution as compared to the microbiome analyses performed in other mouse cohorts. This limitation, combined to the fact that FMT never reach complete colonization of bacterial species in receivers, suggests that performing such additional analyses were unlikely to satisfactorily address the point we raised.

Reviewer #3:

This is a review of a manuscript entitled “The gut microbiota and fermentation-derived branched chain hydroxy acids mediate the health benefits of yogurt consumption in obese mice” authored by Daniel et al. In this manuscript, the authors show that dietary yogurt improves glucose homeostasis and reduces hepatic insulin resistance and liver steatosis in T2D obesity mouse model. The authors also show that yogurt intake alters the liver metabolome particularly increasing branched chain hydroxy acids (BCHA), which correlate with improved metabolic parameters and improve insulin response in glucose metabolism in both liver and muscle tissues.

The manuscript is of great significance, it represents a great contribution to the field and sheds some light on the “dark” area of host-microbiome metabolic interactions. The manuscript is well written, and the data is presented clearly.

Response: We greatly thank the reviewer for her/his positive feedback on the value of our work. Each of the comments and suggestions was carefully considered and answered as detailed below.

Major comments:

1) The authors used several LC-MS Methods for the characterization of the signatures of the different samples. Some samples were analyzed by Metabolon while others were analyzed by the authors. In Methods section “Lipids and Metabolites Profiling” the authors provide different information and description of their processes starting from sample prep to chromatographic gradient and MS settings for the different analyses. This information is very important to be able to replicate the experiments and the authors need to describe their Methods consistently for all their LC-MS analyses and not only a few. Some important parameters are: weight of tissue and volume used for extraction, reconstitution solvent composition, chromatographic gradient, column temperature, QQQ transitions, flow rate, etc. Some of these parameters are described for some Methods but not consistently throughout the Methods section.

Response: We thank the Reviewer and apologize for not reporting the Methods in a detailed and consistent manner as two different contracted research organizations generated these data for us. The samples (liver biopsies, lyophilized yogurt & milk) were analyzed by targeted lipidomics (CLP platform) and untargeted metabolomics (HD4 platform) by Metabolon (Morrisville, North Carolina, USA). The samples (lyophilized yogurt & milk, milk, yogurts, fermented milks and skyr) were analyzed by targeted NMR analysis by Bioaster (Lyon, France). The mouse samples (plasma, liver, muscle, jejunum, colonic content) were analyzed by targeted LC-MS/MS by Bioaster (Lyon, France). BCHA were not quantified by Danone Nutricia Research.

We revised the Methods to report requested details in a consistent manner (see p.24-35).

2) Unlike the metabolomic LC-MS analysis Methods used by Metabolon, the lipidomics analysis Methods were not described at all.

Response: We thank the Reviewer for the comment, and we have now integrated the detailed method for lipidomics CLP analysis in the revised manuscript (see p.24-35).

Minor comments:

1) In page 17 line 417, an “S” is missing in the word States.

Response: We thank Reviewer for the comment. The modification was made in the text.

2) In page 26 line 659, it should say “flushed”.

Response: We thank Reviewer for the comment. The modification was made in the text.

Other modifications made in the documents:

- While working on the manuscript revision, we have noticed in Figure 4E that reporting the hepatic triglycerides as log transformed values may be confusing versus our statistical analysis done on untransformed data. Therefore, we have updated accordingly the Figure 4E y-axis (hepatic triglyceride, $\mu\text{mol/g}$).
- We added the HDCA measure in Fig. S3 (new panel F) to correspond to the sentence “we found an increase of the 6 α -hydroxylated acid, hydoxycholic acid (HDCA) in fecal content

of mice fed yogurt for 3 weeks in Studies 1 and 2 (**Figure 3D**), and this effect was further visually amplified after 12 weeks of yogurt feeding in Study 3 indicating altered intestinal bacteria bile metabolism (**Figure S3F, Table S1**)". See p.8, L187-190.

- In addition, we have noticed a few minor and unfortunate errors in Supplementary Table 1 (mainly copy-paste issues from our internal reports and rounding corrections) that we have now corrected in this new submission (see revised Supp. Table 1). This has no impact on the main text and figures of the paper which were based on the correct statistics.
- A typo mistake was identified in the Figure 3, panel C: Y2vs H1 was corrected by Y2vsH2.
- A copy-paste mistake was identified in the excel file and corrected in the Figure 3, panel E: for the HOMA-IR index, the p-value for H1-T and Y1-T difference is not 0.06 but 0.02, thus significant (\$). The main text was modified accordingly (see p.8, L195-199).
- A mistake in the text about p-values for *Cd36* (p=0.07 instead of 0.06) and *ACACA* (p=0.07 instead of 0.05) genes was corrected (See p.6-7, L153-154).
- Few abbreviations and references for method section were missing and updated.
- QPCR method of hepatic and ileal tissues were previously omitted and added in the Methods section of the revised paper.

REVIEWERS' COMMENTS

Reviewer #1 (Remarks to the Author):

The authors have satisfactorily addressed my comments from the prior review through revisions to the text, tables, figures and rebuttal provided in the "response to reviewers." No further questions from me.

Reviewer #2 (Remarks to the Author):

The authors have done a good job addressing the comments and concerns raised with the first submission.

Reviewer #3 (Remarks to the Author):

The authors have addressed all my concerns in the revisions.